

# Stratospheric intrusion depth and its effect on surface cyclogenesis: An idealized PV inversion experiment

Michael A. Barnes[1,2], Thando Ndarana[2], Michael Sprenger[3] and Willem A. Landman[2]

[1] Marine Research Unit, South African Weather Service, Cape Town, South Africa
[2] Department of Geography, Geoinformatics and Meteorology, University of Pretoria, Pretoria, South Africa
[3] Institute for Atmospheric and Climate Science, ETH Zurich, Zurich, Switzerland

*Correspondence to*: Michael A. Barnes (michael.barnes@weathersa.co.za)

**Abstract.** Stratospheric intrusions of high potential vorticity (PV) air are well-known drivers of cyclonic development
throughout the troposphere. PV anomalies have been well studied with respect to their effect on surface cyclogenesis. A gap
however exists in the scientific literature describing the effect that stratospheric intrusion depth has on the amount of surface
cyclogenetic forcing at the surface. Numerical experiments using PV inversion diagnostics reveal that stratospheric depth is
crucial in the amount of cyclogenesis at the surface. In an idealised setting, shallow intrusions (above 300hPa) resulted in a
marginal effect on the surface, whilst growing stratospheric depth resulted in enhanced surface pressure anomalies and surface
cyclogenetic forcing. The horizontal extent of the intrusion is shown to be more important in developing deeper surface
cyclones than the vertical depth of the stratospheric intrusion. The size of vertical intrusion depths is however an important
factor determining the surface relative vorticity, with larger intrusions resulting in stronger cyclonic circulations. Deeper
stratospheric intrusions also result in intrusions reaching closer to the surface. The proximity of intrusions to the surface is a
crucial factor favouring surface cyclogenetic forcing. This factor is however constrained by the height of the dynamical
tropopause above the surface.

## 1 Introduction

Potential vorticity (PV) has been well established as a highly useful and important parameter within dynamical meteorology
(Hoskins et al. 1985). The usefulness of a PV framework for both operational and academic meteorological analyses is
primarily down to two characteristics of PV. The first is the fact PV is conserved for adiabatic and frictionless flow (Hoskins
et al. 1985; Holton and Hakim 2013). The second of these characteristics is the invertibility of PV (Røsting and Kristjánsson
2012). PV inversion, under suitable balance and boundary conditions, allows for the calculation of other meteorological
parameters such as pressure and wind velocity as a result of a distribution of PV (Davis 1992; Lackmann 2011). Kleinschmidt
(1950) introduced the initial ideas of PV invertibility for specific cases, attributing circulation patterns in the low-levels to an
upper-level PV anomaly and introducing the idea of deducing wind, pressure and temperature fields from PV distributions.
These ideas became more refined and generalised through the development of quasi-geostrophic theory (Charney and Stern




1962) which are still continually being developed and improved on today. PV frameworks and invertibility however only started to be a staple of dynamical meteorological analyses after the landmark paper by Hoskins et al. (1985).

PV invertibility has allowed for the study of different meteorological phenomena such as cyclogenesis from a PV perspective (Davis and Emanuel 1991). A key principle in PV analyses is the definition of the dynamical tropopause. Traditionally, the

tropopause separates the stratosphere, which is highly stratified, from the troposphere (Kunz et al. 2011). The strict definition from a dynamical or PV perspective is the based on the gradient of isentropic PV contours (Reed 1955). However, for simplicity, a PV iso-surface is often used. The exact value of PV often differs, however, -1.5 and -2.0 PVU contours (in the southern hemisphere) are most common (Lackmann 2011). The identification of the dynamical tropopause is crucial in PV analyses. Tropospheric folds can reveal upper-tropospheric fronts and upper-level PV anomalies (Sprenger 2003). High-PV

anomalies of stratospheric air are often introduced into the troposphere by Rossby wave breaking (RWB) (eg. Thorncroft et al. 1993; Ndarana and Waugh 2011; Barnes et al. 2021a). PV inversion shows that these high-PV (negative) anomalies can result in the cyclonic flow around the anomaly and cyclogenesis. Basic theory of high-PV anomalies has been discussed by various authors and basic meteorological texts (Hoskins et al. 1985; Lackmann 2011; Holton and Hakim 2013) and has led to the basic conceptual model for cyclonic PV anomalies as shown in Figure 1. The conceptual model clearly shows the vast

cyclonic motion around the upper-level PV anomaly. This also extends to the surface beneath the upper-level anomaly.

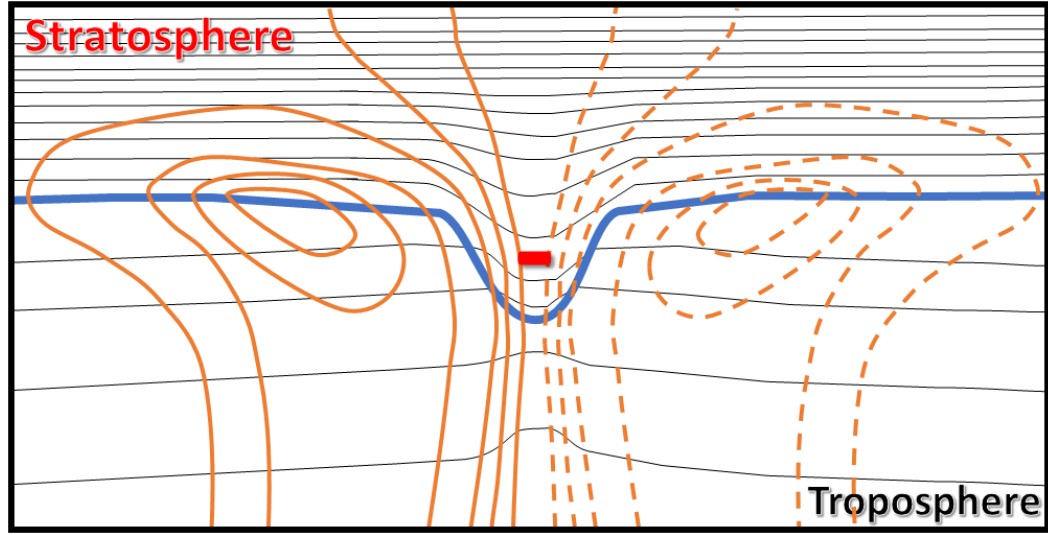

**Figure 1: Conceptual model of a cross-section through a high-PV anomaly (negative sign in red) in the Southern Hemisphere [Adapted from Lackmann (2011)]. Black lines represent isentropes, whilst orange lines represent represent meridional wind velocities (dotted negative, solid positive). The think blue line represents the dynamical tropopause (a constant PV contour).**

PV inversion is the perfect tool to infer how various PV anomalies affect cyclogenesis throughout the troposphere and stratosphere. PV inversion has been used extensively throughout the scientific literature for this purpose. For example, extratropical cyclogenesis has been studied in the context of how different PV anomalies throughout the troposphere interact to influence cyclogenesis (Huo et al. 1999). Here it was shown through case studies that both the vertical alignment and phase





of PV anomalies throughout the troposphere together with interactions between the upper-level anomalies and smaller

anomalies within the upper-level mean flow are important to cyclogenesis. PV inversion has also been used to show the effect of upper-level anomalies in a variety of other extra-tropical (Ahmadi-Givi et al. 2004; Pang and Fu 2017) and tropical (Moller and Montgomery 2000; Arakane and Hsu 2020) settings. Other studies have used PV inversion to diagnose numerical weather prediction (NWP) errors (Brennan and Lackmann 2005) and the effect on downstream precipitation (Baxter et al. 2011). Few studies have used PV inversion in an idealised setting to study the upper-level and stratospheric intrusion depth's influence on

cyclogenesis, especially from a Southern Hemispheric point of view. A study of stratospheric depths in relation to surface cyclogenesis was performed in relation to cut-off lows by Barnes et al. (2021a). This study was done from a climatological point of view in the Southern Hemisphere. The results show that stratospheric intrusions with a -1.5PVU stratospheric tropopause associated with 250hPa cut-off lows that extend to 300hPa or below, are more likely to result in surface cyclogenesis. This COL extension climatology by Barnes et al. (2021a) was based on real-case reanalysis data. This implies

that there are other processes such as low-level PV anomalies that influence the amount of cyclogenesis occurring at the surface.

In this study, the effect that the stratospheric intrusion depth has on surface cyclogenetic forcing is studied in an idealised setting. Although the effect that stratospheric intrusions has on surface cyclogenesis is not a new concept (eg. Hoskins et al., 1985), this study examines the effect that stratospheric depth and intrusion characteristics has on surface cyclogenetic forcing

in a highly systematic way. This allows for a correlation between intrusion depths, widths, and intensities with the amount of cyclogenetic forcing at the surface. Additionally, little scientific literature involving stratospheric intrusions depths in idealised settings, has been performed in a Southern Hemisphere where PV is negative. This study looks at these idealised PV intrusions into the troposphere in a highly systematic way. This study aims to enhance our understanding of the effect PV intrusion depth has on surface cyclogenetic forcing as described in basic theoretical texts (eg. Hoskins et al. 1985). The idealised simulations

also confirm and corroborate the findings and hypothesis of the climatology by Barnes et al. (2021a) that deeper intrusions are responsible for deeper COLs and surface cyclone development. A collection of numerical experiments using the power of PV inversion is used. Various experiments using variations in the depth and intensity of the simulated intrusion as well as variations in the dynamical tropopause height are performed. This paper is organised as follows. Section 2 introduces the piecewise PV inversion algorithms used for the experiment as well as the various experimental setups for each test. The results of these tests

are presented in Section 3. The results are finally discussed holistically and conclusions are drawn in Section 4.

## 2 Methodology

### 2.1 Piecewise PV inversion algorithm

PV invertibility is a mathematical construct. The basic mathematical ideas have been fully described in many textbooks. From Holton and Hakim (2013), quasi-geostrophic PV ($q$) can be expressed mathematically by:





$$q = \zeta_g + f + f\frac{\partial}{\partial z}\left(\frac{\partial \bar{\theta}}{\partial z}^{-1}\theta\right) \qquad (1)$$

with $\zeta_g$ the geostrophic relative vorticity, $f$ the Coriolis parameter ($f < 0$ in the Southern Hemisphere), $\theta$ the potential

temperature and $\bar{\theta}$ is the potential temperature of the reference state. The aim of the invertibility principle is to return a variable,

say pressure $p$ by integrating (1).

One of the more popular methodologies for solving the equation (1) is the piecewise PV inversion technique. Various variations

of this technique are tested and discussed by Davis (1992). It was found that the various techniques to solve the problem in a

piecewise approach were found to produce only small differences near the anomalies studied.

In this study, numerical experiments are performed utilising the PV inversion framework of Fehlmann (1997). The set of code

provides a diagnostic for reanalysis datasets to diagnose the effect that PV anomalies have on the surrounding meteorological

parameters. In this study, however, we use an extension of these algorithms which allows for more idealised experimentation

(Fehlmann 1997; Sprenger 2007). The set of numerical codes solve the Neumann boundary problem for potential vorticity $q$

and the streamfunction $\psi$ from which the wind components can be derived given by the quasi-geostrophic PV

$$q = \frac{\partial^2 \psi}{\partial x^2} + \frac{\partial^2 \psi}{\partial y^2} + \frac{f^2}{\bar{\rho}}\frac{\partial}{\partial z}\left(\frac{\bar{\rho}}{\bar{N}^2}\cdot\frac{\partial \psi}{\partial z}\right) \qquad (2)$$

where $\bar{\rho}$ and $\bar{N}$ denote the density and Brunt-Väsälä for the reference state respectively. The boundary values of potential

temperature at the lower and upper boundaries are given by:

$$g\cdot\frac{\theta^*}{\bar{\theta}} = f\cdot\frac{\partial \psi}{\partial z} \qquad (3)$$

whilst the lateral boundary condition for the u and v wind components are given by

$$u = \frac{\partial \psi}{\partial y} \; ; v = \frac{\partial \psi}{\partial x} \qquad (4)$$

Using various partial differential equations and discretisation techniques as shown in detail by Sprenger (2007) and Fehlmann

(1997), the above problem can be solved numerically using a piecewise numerical approach. For details about the numerical

aspects of the PV inversion framework, based on successive over-relaxations, see Fehlmann (1997) and Sprenger (2007).

The idealised setup tool of Fehlmann (1997) allows the user to create an idealised basic state. This basic state is based on a

user-defined jet stream (height, width, depth and intensity), dynamical tropopause height, static stability (of both the

troposphere and stratosphere), latitude and surface baroclinicity. Potential temperature profiles are constructed from the

available input by so-called "kink" functions (Fehlmann 1997). Once defined and setup, a PV anomaly can be defined and

introduced into the basic state. The code allows the user to define the intensity of the anomaly, vertical and horizontal

dimensions and positions.





## 2.2 Experimental setup

The idealised numerical experimental domain in this study has a zonal dimension of 7500km and a meridional dimension of
5000km with a 25km horizontal resolution. In the vertical, 200 levels are specified with the upper limit at 20000m above
ground level (AGL) and the lower limit on the surface. The vertical levels have a resolution of 100m. The PV inversion
algorithm allows the user to specify the surrounding environment for the experiment. In this study, we aim to replicate the
conditions of the climatology presented by Barnes et al. (2021a), where the dynamical tropopause is considered to be the -1.5
PVU iso-surface. The PV inversion algorithm however considers the tropopause height based on a higher PV contour given a
specific height AGL with respect to the static stability parameters. To comply with the convention of the code, the dynamical
tropopause is set at a specific height value AGL. The height of the -1.5 PVU iso-surface is calculated from the field after setup.
The static stability parameters are then set in such a way that the -1.5 PVU iso-surface can be considered as the clear divide
between the stratosphere and the troposphere. This is shown by the meridional PV cross-section of the basic state in Figure 2.
In this field, the dynamical tropopause was set to 12500m AGL whilst the resulting -1.5 PVU contour (effective dynamical
tropopause) calculated to be at 11285m AGL.

The algorithm also allows for the specification of the jet stream in the upper-levels. The jet was centred around the specified
dynamical tropopause with a 4000m stratospheric depth and 6000m tropospheric depth (Figure 3). The westerly jet stream is
specified to be zonal with the horizontal centre of the jet in the centre of the domain and a maximum velocity of $35\text{m.s}^{-1}$. Figure
3 (left) shows the zonal wind speed at the height of the specified dynamical tropopause. The Coriolis force is applied using a
constant $f$-plane approximation. For the entirety of this study, this was deemed to be 42ºS. From the above parameters, the
algorithm calculates all the basic state meteorological variables. The upper-level pressure field just below the dynamical
tropopause and -1.5 PVU contour (at 10000m AGL) that results from the preparation algorithm is shown in Figure 3 (right).
No meridional flow exists throughout the basic state domain. Additionally, it is pertinent to point out that the surface field is
setup in such a way that no baroclinicity is present. The lack of baroclinicity results in a surface of a constant pressure of
1000hPa and no surface wind flow. The resulting environment from the above and as shown in Figure 2 and Figure 3 is deemed
to be the basic state for this study. Except for the specified dynamical tropopause height, this remains unchanged throughout
the study.





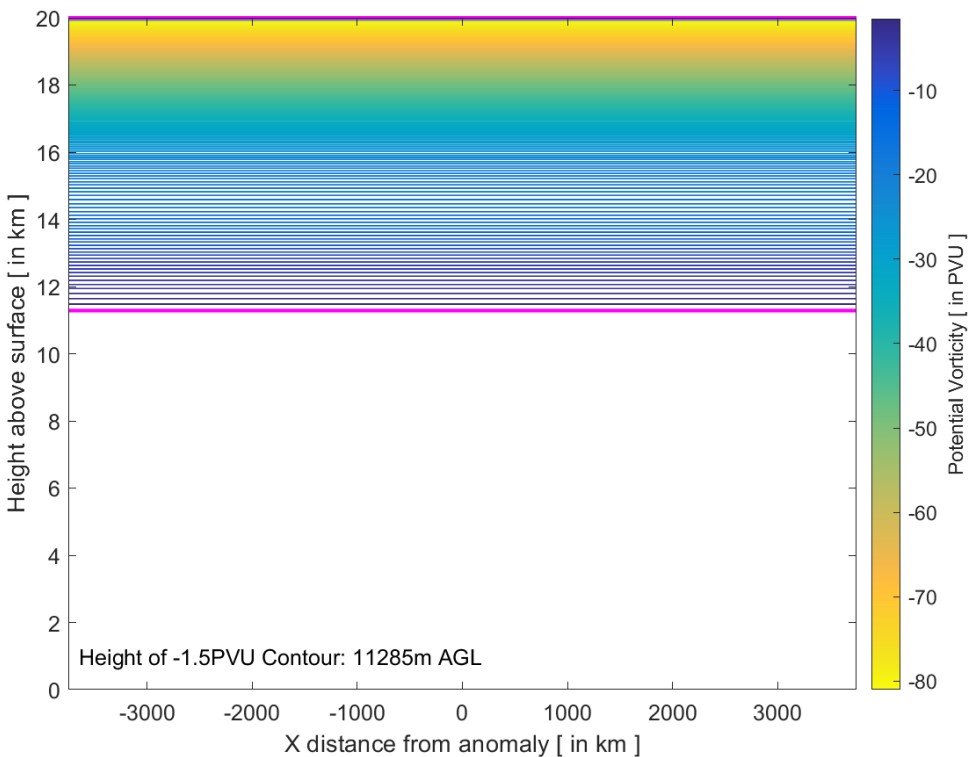

**Figure 2: PV zonal cross-section through the centre of the domain. Tropopause was specified at a height of 12500m AGL. The -1.5 PVU contour (highlighted in a thick magenta line) was calculated to be 11285m AGL.**

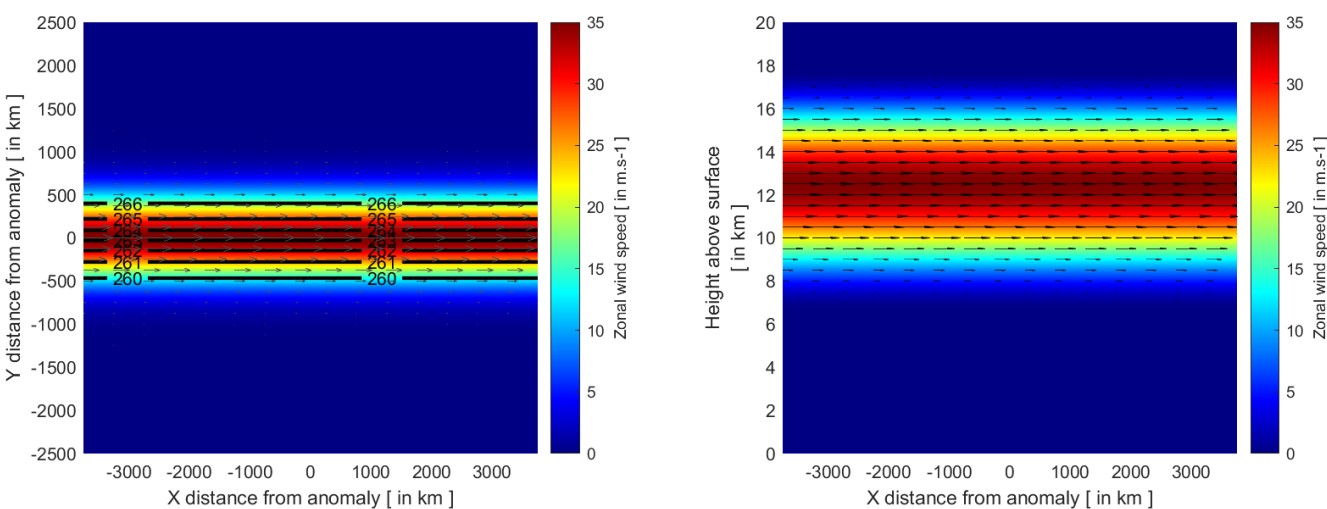

**Figure 3: The jet stream of the model setup.**
**Left: The zonal wind speed of the jet stream at the height of given model tropopause (12500m AGL). Pressure contours (in hPa) are overlaid together with zonal wind quivers.**
**Right: A cross-sectional view of the zonal wind associated with the jet stream through the centre of the domain overlaid with zonal wind quivers.**





The study examines how the meteorological fields changed based on a high-PV anomaly which is forced into the domain. The anomaly is a three-dimensional PV anomaly given by:

$$ANO = min\left\{-1.5, -\left(4 \times \left[e^{\frac{-(z-z_{pos})}{z_{size}}}\right] \times \left[e^{\frac{-(y-y_{pos})}{y_{size}}}\right] \times \left[e^{\frac{-(x-x_i)}{x_{size}}}\right]\right)\right\} \tag{5}$$

where $x, y, z$ are the horizontal and vertical coordinates, $x_{pos}, y_{pos}, z_{pos}$ are the $x, y, z$ coordinates of the centre of the anomaly and $x_{size}, y_{size}, z_{size}$ are the horizontal and vertical radial dimensions of the anomaly. For this study, the anomaly magnitude is set at standard value -1.5 PVU. Equation (5) results in an anomaly with a minimum PV intensity of -1.5 PVU and which increases outward from the central minimum for a distance $x_{size}, y_{size}, z_{size}$. An example of this anomaly is shown in the horizontal and vertical profiles in Figure 4. The anomaly shown in Figure 4 has a $x_{size}, y_{size} = 200km$ resulting in a total

maximum width of 400km. The specified $z_{size} = 5000m$ results in a total vertical size of 10000km. This excludes a "halo" of increasing values around the specified anomaly. Applying an anomaly such as shown in Figure 4, results in a lowering of the values in the stratosphere. More importantly, the anomaly results in a tongue of high-PV values emerging below the -1.5 PVU contour.



**Figure 4: Example of a PV anomaly forced into the idealised domain by means of a longitudinal cross section (A) and a horizontal cross-section (B) through the centre of the anomaly. The anomaly has a maximum horizontal width along the minor axis of 400km and a height of 10000m. This excludes the "halo" of decreasing values to zero around it. The red line is defined as the anomaly radial width ($x_{size}$, $y_{size}$), whilst the grey line is defined as the anomaly radial height ($z_{size}$). The anomaly magnitude (in this case -1.5 PVU) is shown by the magenta contour.**






The control experiment (Experiment 0) is used as a reference to gain an understanding of what the algorithm produces and how the PV anomaly affects the meteorological parameters within the domain. Experiment 0 uses the specified values above, namely a dynamical tropopause height of 12500m, an anomaly radial width of 200km and an anomaly radial height of 5000m. The meteorological changes that occur because of the introduction of the anomaly are then analysed, with special focus on surface cyclogenetic forcing as measured by the induced surface relative vorticity and surface pressure changes. This is

discussed in the context of basic PV theoretical concepts. Further, we test the effect of changes to four different parameters with respect to the anomaly and tropopause and their effect on the surface cyclogenetic forcing.

The first (Experiment 1) experiment systemically explores the effect of the depth of stratospheric intrusions on the amount of surface cyclogenetic forcing. The effect that depth of the stratospheric intrusion has on surface cyclogenetic forcing is tested by varying the anomaly radial height. Experiments are performed with anomaly radial height values of lower (2500m) and

higher (7500m and 10000m) than the control experiment (Experiment 0, 5000m). Experiment 1 is performed with a constant dynamical tropopause height of 12500m and a constant anomaly radial width of 200km. The varied anomaly radial heights with a constant dynamical tropopause results in tongues of high-PV air extending further towards the surface, as observed in stratospheric intrusions and tropopause fold.

Secondly, the effect that the height of the dynamical tropopause above the surface has on surface cyclogenetic forcing is

explored in Experiment 2. The second experiment comprises of three separate model experiments with varying dynamical tropopause heights with constant anomaly radial height of 5000m and a constant anomaly radial width of 200km. Dynamical tropopause heights of 15000m and 10000m are used and the results compared to the control experiment (Experiment 0, 12500m). This experiment gives us an indication of whether the depth of the stratospheric intrusion is more important than the proximity of the intrusion to the surface. This notion was hypothesised in Barnes et al. (2021a).

The third set of experiments, Experiment 3, reasserts the notion of stratospheric depth versus proximity to the ground. In Experiment 3, only the anomaly radial width is kept constant at 200km. Both the dynamical tropopause height as well as the anomaly radial height are varied simultaneously such that the eventual height of the intrusion AGL is similar. In this experiment we use dynamical tropopause heights of 15000m and 10000m. Testing anomaly radial heights in 500m intervals, we compare experiments that result in the closest stratospheric intrusion height AGL compared to that of Experiment 0.

Experiment 4 experiments with the magnitude of the intruding anomaly and test how it affects cyclogenetic forcing at the surface. For this experiment, the anomaly description remains the same as in (5). However, all values that are greater than the specified anomaly magnitude are assigned a value equivalent to the anomaly magnitude. A higher (-2 PVU) and lower scenario (-1 PVU) are tested and compared to the control experiment (Experiment 0, -1.5 PVU).

Finally, the effect that the horizontal size of stratospheric intrusions has on surface cyclogenetic forcing is tested by varying

the anomaly radial width (Experiment 5). Tests with anomaly radial width values of 100m, 200m and 400m are performed with a constant dynamical tropopause height of 12500m and a constant anomaly radial height of 5000m. All the above experiments are also provided in the flow chart shown in Figure 5.





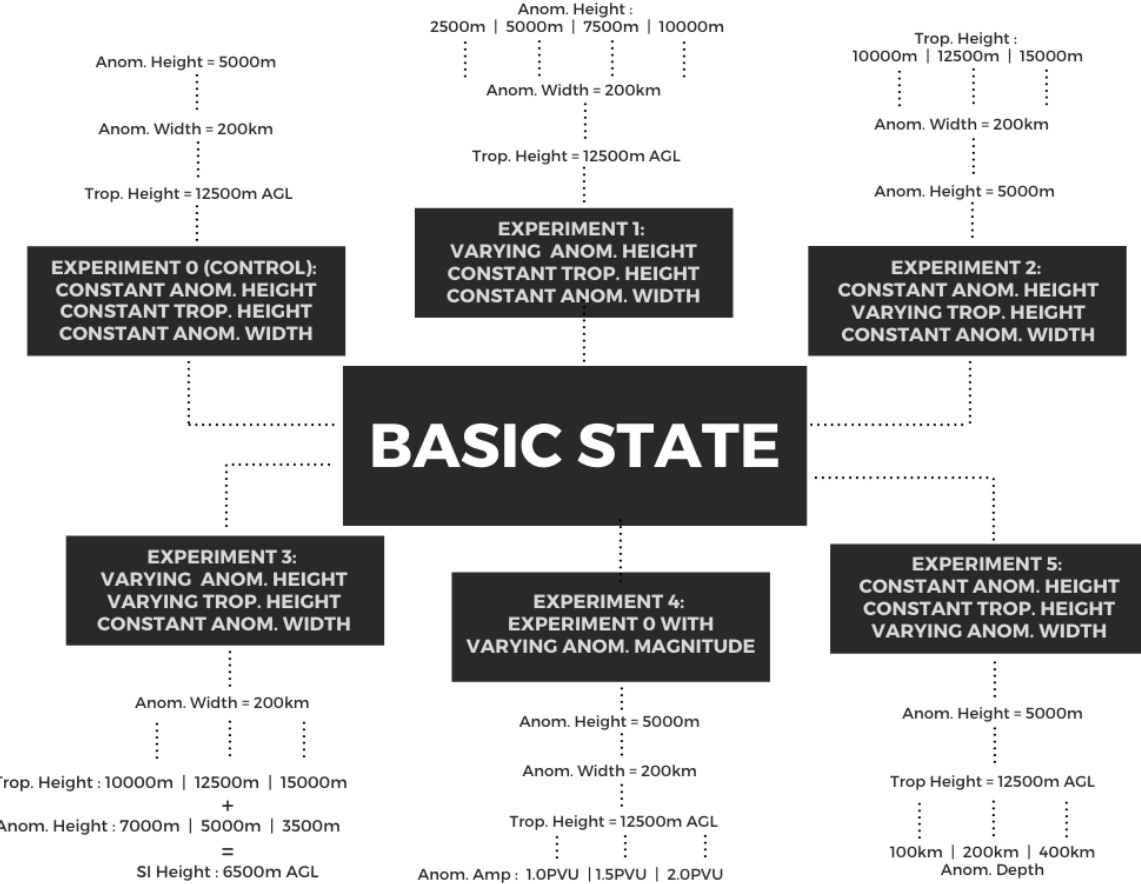

**Figure 5: Flow chart of all experiments performed using the PV inversion algorithm. In each experiment, the basic state remains**
**the same with varied dimensions of the PV anomaly, magnitude of the PV anomaly and the height of the dynamical tropopause**
**AGL.**

## 2.3 Definition of cyclogenesis to describe cyclogenetic forcing

Cyclone development has long been studied throughout the troposphere. The key to studying these low-pressure systems is the creation of algorithms to track them. Tracking algorithms provide guidance on the parameters that constitute the presence of a surface cyclone. Different tracking algorithms use different parameters for the tracking of different characteristics of surface cyclones (Neu et al. 2013). Surface cyclones have been identified by MSLP minima using various thresholds associated for central depths and pressure gradients (Blender and Schubert 2000; Nissen et al. 2010). Other algorithms use surface relative vorticity ($\zeta$) as a measure of cyclogenesis (eg. Flaounas et al. 2013, 2014; Hofstätter et al. 2016). Relative vorticity thresholds within these studies, although predominantly based in the European region, remain relatively consistent. Relative vorticities in the order of $10^{-5}s^{-1}$ are generally used for this purpose. Flaounas et al. (2013) used the requirement of $10^{-5}s^{-1}$ but with the addition that at least one grid-cell as a part of the event greater than $8 \times 10^{-5}s^{-1}$, which takes into account a threshold found in mature surface cyclones. Others, are more consistent with cyclogenesis requiring a value of around $3 \times 10^{-5}s^{-1}$



(Flaounas et al. 2014; Hofstätter et al. 2016). In this study, we use the threshold of $-3 \times 10^{-5} s^{-1}$ (negative as we are dealing with southern hemispheric values) as the value of cyclogenesis. Cyclogenesis is however a process which implies a temporal aspect. However, this study is performed in a stationary context. Thus the values cyclogenesis definitions are used here as a guide to the extent of cyclogenetic forcing that is supplied by the upper-level PV structure onto the surface. In accordance with the scientific literature (eg. Flaounas et al. 2013, 2014; Hofstätter et al. 2016), we stipulate the following definitions:

    1)   Cyclogenesis: $\zeta \leq -3 \times 10^{-5} s^{-1}$

    2)   Mature cyclogenesis: $\zeta \leq -8 \times 10^{-5} s^{-1}$

    3)   Weak cyclogenesis: $\zeta \approx -10^{-5} s^{-1}$, $\zeta > -3 \times 10^{-5} s^{-1}$

Relative vorticity is analysed in conjunction with surface pressure deepening. In this regard, a lowering of the surface pressure and the development of a surface pressure minimum is required. A cyclone can only be said to be developed if the central pressure is enclosed by a closed isohypse at a contour interval of 1hPa. The amount of surface pressure deepening will be analysed and discussed in the context of the type of cyclogenesis (cyclogenetic forcing) identified as defined above. Within the confines of this study, the relative vorticity and surface pressure conditions need to be met for cyclogenesis to have been deemed to have occurred. It should be noted that cyclogenesis in general is a development process and as such has a temporal aspect to it. For the confines of this study, we use these above definitions of when cyclogenesis occurs to describe the type and "strength" of surface cyclogenetic forcing in the static environment of the numerical experiment. If relative vorticity values at the surface fall outside of the definitions above, it is said that cyclogenetic forcing is small or negligible.

## 3 Results

### 3.1 Experiment 0: An idealised stratospheric intrusion and its effect on the domain

A basic, reference experiment is performed to reconstruct the conceptual model of a PV anomaly that extends from the stratosphere. Figure 6 shows a stratospheric intrusion simulated to a depth of 5000m from the dynamical tropopause stipulated at 12500m AGL. The stratospheric intrusion has the standard horizontal radial width of 200km. The -1.5 PVU contour is calculated to be at a height of 11287m AGL (as described in Section 2.2). The stratospheric intrusion extends to a depth of 6594m AGL. Figure 6 also reveals cyclonic motion that is induced around the stratospheric intrusion as is seen in the conceptual model in Figure 1. The cyclonic development is shown in Figure 6 by the positive meridional wind velocities (wind flow "into the page") shown by the solid grey contours to the west of the intrusion and the negative meridional wind velocities (wind flow "out of the page") shown by the dashed grey contours. The upper-level cyclonic development emerges in the upper-level pressure fields as an amplifying trough as shown in Figure 7A. With time, the continued cyclonic development in the upper-levels could result in a closed circulation or COL. This re-emphasises the effect of basic PV theory that shows that cut-off lows are generated from high-PV intrusions of stratospheric air as shown by Hoskins et al. (1985). Although strong cyclonic





rotation is confined to the area around the intrusion, weak cyclonic rotation is present throughout much of the cross-sectional domain, including the surface. This is shown by the outer-most wind velocity contours in Figure 6.

Figure 7B shows the surface pressure isobars (black contours) together with the surface wind vectors. Before the introduction of the PV anomaly, this field is set at a constant 1000hPa. It is clear that the stratospheric intrusion has resulted in a decrease in the surface pressure and has induced cyclonic rotation around the axis of the stratospheric intrusion, as predicted by theory. A drop of 3hPa in surface pressure is observed as a result of the introduction of the stratospheric intrusion. Moreover, this low-pressure minimum is enclosed by at least one isohypse. Relative vorticities within the centre of the surface circulation are

shown by shaded colours in Figure 7B. The lowest relative vorticity observed within the induced surface circulation is $0.9 \times 10^{-5} s^{-1}$, which falls barely outside the thresholds for surface cyclogenesis as defined in Section 2.3. Although by this definition the surface cyclogenetic forcing is very small, this is very close (within $1 \times 10^{-5} s^{-1}$) to the lower limit for weak surface cyclogenetic forcing.

A similar intrusion was observed in the South Atlantic that resulted in a similar decrease in surface pressure and the

development of a surface cyclone (Iwabe and Da Rocha 2009). In that observational study, a similar pressure decrease was seen with the central surface pressure of the surface cyclone decreasing by 4hPa within six hours.

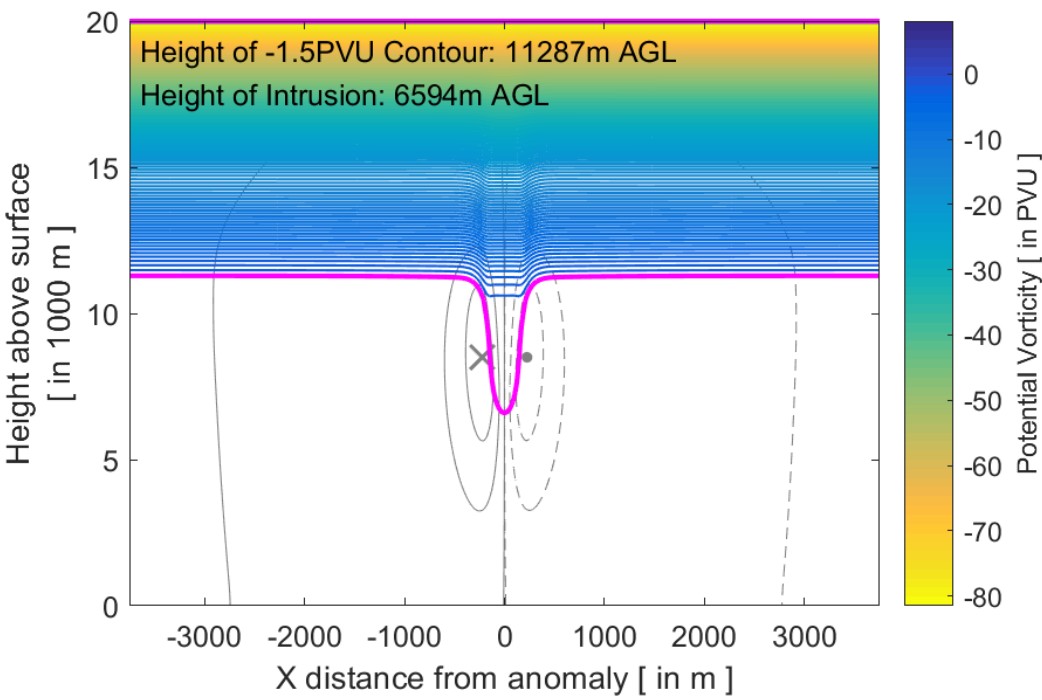

**Figure 6: Stratospheric intrusion with a radial width of 200km and depth of 5000m from the dynamical tropopause specified at 12500m AGL. Meridional wind velocities are shown by the grey contours. Solid grey contours and the "X" indicate winds moving**
**into the page, whilst dashed grey contours and the "Dot" indicate winds coming out of the page.**

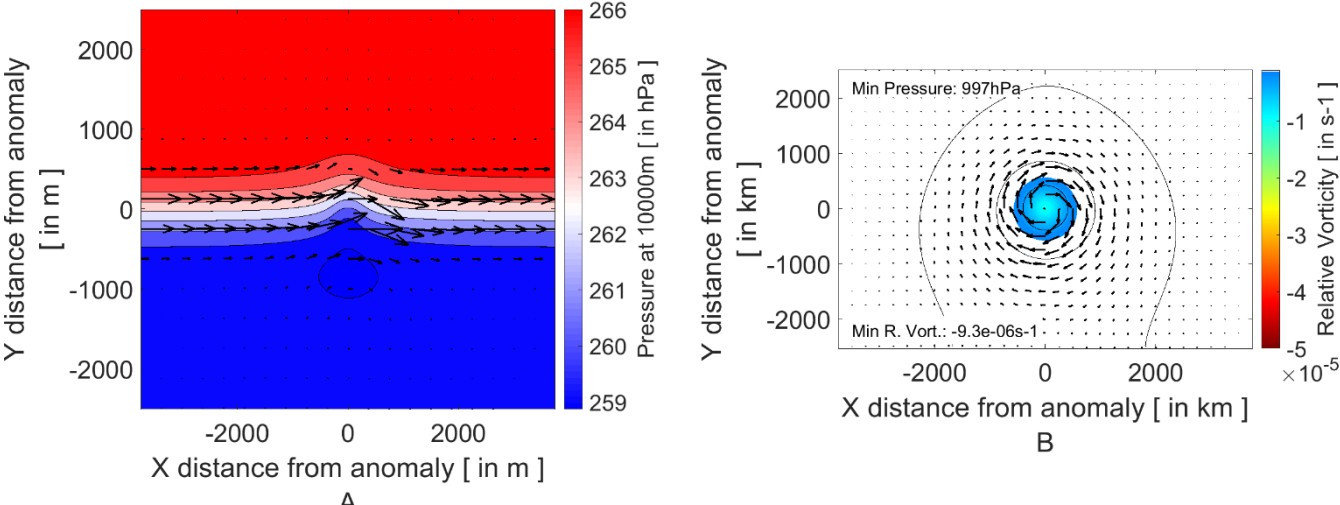

**Figure 7: A: Upper-level (10000m AGL, 2500m below the experimentally defined dynamical tropopause) pressure field (shaded) together with wind vectors plotted as black arrows.**
**B: Surface pressure isobars (black lines) and surface relative vorticity (shaded) together with surface wind vectors plotted as black arrows.**

### 3.2 Experiment 1: Varying stratospheric intrusion depth

The effect that the depth of the stratospheric intrusions has on surface cyclogenetic forcing is investigated by systemically varying the anomaly radial depth (as defined in Section 2.2) of Experiment 0. The results of a selection of these varying stratospheric intrusion depth experiments are shown in Figure 8 and a summary of the full set of experiments are shown in Figure 9. Varying stratospheric intrusion depths are all compared to the control experiment as shown in Experiment 0. For ease of reference and comparison we show this experiment again in Figure 8-A2 and Figure 8-B2 and is highlighted in green in Figure 9.

Using shallower intrusion depths than in Experiment 0 resulted in limited and weaker cyclonic rotation around the stratospheric intrusion as shown in the shallow intrusion experiment in Figure 8-A1. Maximum meridional wind velocities are in fact almost half of that of Experiment 0 using half the anomaly depth (6m.s$^{-1}$ in Figure 8-A1 and B1 compared to 11m.s$^{-1}$ in Figure 8-A2 and B2). The sphere of influence of the shallower intrusion on the surrounding troposphere is much less than with a standard value of 5000m (Experiment 0, Figure 8-A2 and B2). In comparison, tropospheric intrusions to greater depths resulted in far greater cyclogenetic forcing around the anomaly (as depicted in Figure 8-A3 and Figure 8-A4). Maximum meridional wind velocities of the deeper examples in Figure 8 in fact strengthened to 17m.s$^{-1}$ (Figure 8-A3) and 22m.s$^{-1}$ (Figure 8-A3) using 7500m and 10000m anomaly depths respectively. The enhanced cyclogenetic forcing is also shown in Figure 9 by the increase in the minimum cross-section relative vorticity (dashed lines). This increases almost linearly with a constant increase in PV anomaly depth. The growth in mid-level cyclogenetic forcing is however stunted as it gets closer to the surface. Minimum cross-sectional relative vorticities grow much slower with anomaly depth greater than 8000m. Another important feature seen





in these experiments is the location of the maximum mid-tropospheric cyclogenetic forcing. An extended stratospheric
intrusion results in the zone of maximum mid-tropospheric cyclogenetic forcing moving closer to the surface.

As expected, shallower intrusions (less than the 5000m used in Experiment 0) result in a much weaker circulation occurring
on the surface compared to Experiment 0. MSLP and relative vorticities both increase as the depth of the stratospheric intrusion
decreases. Surface pressure decreases below 1hPa according to our definitions of cyclogenetic forcing are deemed to not result
in a closed isohypse. Anomaly radial heights of shallower than 3500m did not meet this criterion. An example of the surface
pressure anomaly for a shallower intrusion is shown in Figure 8-B2. Notably, no closed isohypse at a contour interval of 1hPa
can be seen around the centre of the developing low-pressure minimum. An important factor in our definition of cyclogenetic
forcing is the surface relative vorticity. Figure 9 shows that for intrusions shallower than control Experiment 0, our threshold
of cyclogenesis of $10^{-5}s^{-1}$ is not met (by a factor of $7\times10^{-5}s^{-1}$).

Increasing the depth of the high-PV anomaly also resulted in an increased lowering in the surface pressure and increased
rotation at the surface (Figure 9). The centre of the induced surface pressure anomaly decreases exponentially with increasing
intrusion depth. The 7500m and 10000m anomaly radial depths (depicted in Figure 8-3 and 4) induce an 8hPa and 12hPa
decrease (compared to 3hPa in Experiment 0) in their associated surface pressure respectively. The observational study of
*Cape Storm* in Barnes et al. (2021b) showed a decrease of 6hPa on 7 June 2017 collocated with a stratospheric intrusion to the
550hPa level. The intrusion, similar to that shown in Figure 8-A3, results in a similar surface pressure decrease.

The enhanced cyclonic circulation is also depicted through the increasing relative vorticity present at the surface with increased
PV anomaly depth. Intrusions deeper than Experiment 0, result in weak cyclogenetic forcing with intrusions deeper than 6500m
meet our cyclogenetic forcing criteria. None of the experiments conducted resulted in instantaneous strong cyclogenetic forcing
at the surface. It should be noted that this study does not take any temporal evolution into account as PV inversion is an
instantaneous framework. With increased and continual development of the surface cyclone, the surface cyclone may, with
time, develop into a mature surface cyclone in which mature cyclogenesis exists at the surface. Additionally, in Experiment 0,
an upper-level trough develops coincident with the PV anomaly. With time and continual cyclogenetic forcing in the upper-
levels, under the correct conditions, the upper-level trough may develop its own closed, cyclonic circulation (or COL). This
echoes the findings of Barnes et al. (2021a) who show that, in a climatological sense, deeper intrusions are associated with
COLs that extend to the surface. Despite the lack of temporal aspect, within this experimental framework, surface cyclogenetic
forcing occurs if the depth surpasses the control experiment depth with an intrusion depth of 5000m (6594m AGL) from the
experimental tropopause height of 12500m (11287m -1.5 PVU contour height) AGL.



**Figure 8: A: Longitudinal PV cross-sections through the centre of the forced anomaly. The -1.5 PVU contour (our definition of the dynamical tropopause for this study) is highlighted in by a thick magenta line. Meridional wind velocities are shown by grey contours. Positive velocities (into the page) are represented by solid contours whilst negative velocities are represented by dashed contours (out of the page).**
**B: The effect of the intrusion on the surface pressure and relative vorticity are shown in the right panels. Pressure isobars at a 1hPa contour interval are shown by black lines, whilst relative vorticity is shown by the shading.**
**The panels in rows 1-4 represent different varying stratospheric depths introduced into the domain. For this experiment (Experiment 1), radial anomaly depths given to the system are 2500m (row 1), 5000m (row 2), 7500m (row 3) and 10000m (row 4).**



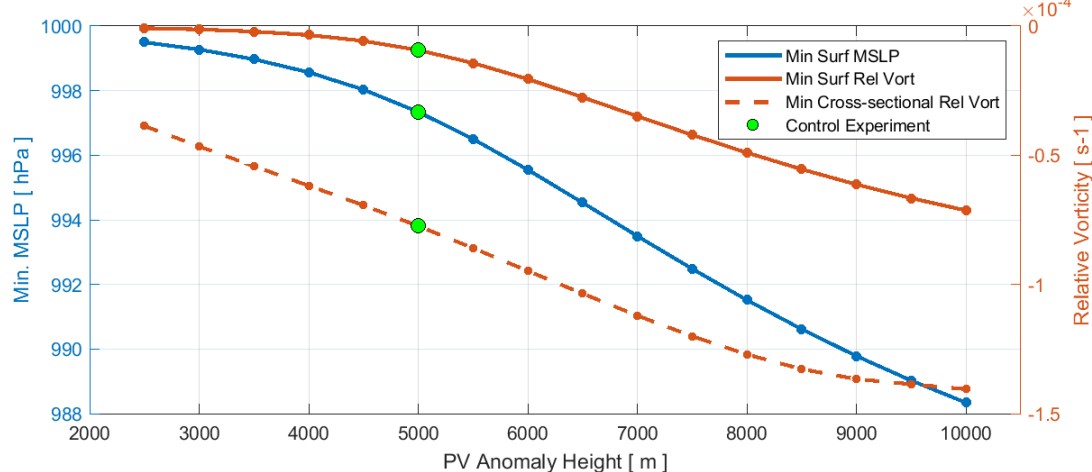

**Figure 9: Changes to surface parameters (solid lines) as a function of anomaly height (depth of intrusion). The minimum MSLP (blue) and relative vorticity (orange) on the surface pressure level are recorded and plotted. The cross-sectional minimum relative vorticity as a function of anomaly depth is also shown by the dashed line. The results in Experiment 0 are highlighted in green for convenience.**

### 3.3 Experiment 2: Varying tropopause height with constant intrusion depth

The height of the tropopause is variable both spatially and seasonally (Kunz et al. 2011). In the summer months the meteorological equator is situated south of the geographical equator resulting in a raised tropopause, with the converse being true for the winter months. Additionally, the temperature differences between equator and the poles result in the dynamical tropopause being situated closer to the surface in the higher latitudes compared to the lower latitudes (Kunz et al. 2011). Barnes et al. (2021a) showed a distinct seasonality and latitudinal discrepancy in the number of COLs that extend to the surface, linking this variability to the height of the tropopause. Lower tropopausal conditions (as found in winter and closer to the poles) tend to produce a greater number of COL extensions in the Southern Hemisphere than when the tropopause is further from the surface (in the summer months and closer to the equator).

The dependence of tropopausal height on surface cyclogenetic forcing is explored here in a systemic and idealized way by changing the height of the specified tropopause with a constant anomaly radial height of 5000m. The constant anomaly radial height simulates stratospheric intrusions of similar vertical "intensity" in different tropopause height regimes. The standard 12500m tropopause height AGL (as seen in Experiment 0) are depicted by a higher scenario (tropopause height of 15000m) and a lower scenario (tropopause height of 10000m) in Figure 10. A full set of experiments comprising of various higher and lower tropopause scenarios and the resulting amount of cyclogenetic forcing are shown in Figure 11. It should be noted that the tropopause height stipulated and the resulting -1.5 PVU contours (which denotes our definition of the dynamical tropopause) do differ. The actual dynamical tropopause heights (-1.5 PVU contours) for the mid (Experiment 0), high (15000m tropopause) and low scenarios (10000m tropopause) are situated at 11287m, 13384m and 9172m AGL.





Figure 10A depicts the PV intrusions in the mid, high and low tropopause scenarios. The decreasing height of the dynamical tropopause results in the stratospheric intrusions effectively being in closer proximity to the surface but with no change in the "intensity" of the intrusion. The difference in tropopause height results in different heights of the associated stratospheric intrusion AGL reaching 6594m (Experiment 0), 8776m (higher scenario) and 4332m AGL (lower scenario) respectively. The resulting effects on the surrounding troposphere and surface are shown in Figure 10 and Figure 11.

As with Experiment 1, Figure 10A shows the differences in the induced cyclonic circulation around the PV anomaly in the upper-levels. The increasing dynamical tropopause height, even with a similar vertical PV intrusion intensity or size, has a similar effect on the circulation around the anomaly as seen in Experiment 1. The enhanced mid-tropospheric circulation is readily seen in the decreasing minimum cross-sectional relative vorticities from the highest tropopause to the lowest tropopause scenario (Figure 11). Minimum cross-sectional relative vorticities show a decrease of about $4 \times 10^{-5} s^{-1}$ in the minimum cross-
sectional relative vorticity between the high and low tropopause height scenarios. Maximum meridional velocities increase from $9 m.s^{-1}$ in the highest tropopause scenario (Figure 10-A1) to $12 m.s^{-1}$ in the control (Experiment 0) or intermediate scenario (Figure 10-A2). The lowest tropopause scenario (Figure 10-A3) contains a maximum meridional velocity of $15 m.s^{-1}$. The increased rotation from one scenario to the next is not a function of the intensity of the anomaly since the amplitude of the anomaly is kept constant in Experiment 2. With no difference in the PV environment, it follows from (1) that

$$\zeta_g + f = q - f \frac{\partial}{\partial z}\left(\frac{\partial \bar{\theta}}{\partial z}^{-1} \theta\right) \qquad (6)$$

A dynamical tropopause situated closer to the surface is also found on a higher-pressure contour compared to that of the dynamical tropopause situated further away from to the surface. From the high, intermediate and low tropopause scenarios in Figure 10, the dynamical tropopause is situated at around the 110hPa, 175hPa and 260hPa pressure levels respectively. Crucially, the dynamical tropopause coincides with a potential temperature in the range of 330-350K for all three scenarios. These factors result in a more tightly packed potential temperature gradient in the tropopause (high static stability) in the
scenario where the dynamical tropopause is closer to the surface compared to the higher tropopause scenario. Therefore, for a PV (q) intrusion of the same intensity, the increased static stability resulting from the lower dynamical tropopause will result in a decrease in the relative vorticity value on the left hand side of (6). Since we are dealing with negative vorticity in and around the anomaly, the decrease in relative vorticity corresponds to increased rotation around the PV anomaly. The converse argument can of course be made for the scenario in which the tropopause is situated further away from the surface, decreasing
static stability and decreasing the cyclonic motion around the anomaly as a result.

Figure 11 shows that the amount of cyclogenetic forcing increases exponentially at the surface with decreased tropopause height. This is seen by both the exponential decrease in pressure and relative vorticity at the surface.

The intermediate scenario (Experiment 0) with an intrusion depth of 6594m resulted in a 3hPa drop in surface pressure. In contrast, with the same vertical intrusion depth, the intrusions from the higher dynamical tropopause (Figure 10-A1 and A2)
resulted in a meagre 1hPa decrease in the surface pressure with a closed isohypse. Relative vorticities within the pressure





minimum however were small in the order of $-10^{-6}s^{-1}$. The low value of relative vorticity is below the threshold for cyclogenesis as outlined in Section 2.4. Therefore, negligible cyclogenetic forcing resulted beneath the stratospheric intrusion from the high dynamical tropopause. A stark contrast is seen in the scenario from the dynamical tropopause situated closer to the surface (Figure 10-A3 and A3). The lower scenario resulted in a doubling of the pressure decrease at the surface (6hPa) compared to

Experiment 0 (3hPa). Additionally, enhanced cyclonic circulation is induced at the surface in the lower scenario. The minimum relative vorticity of $-3\times10^{-5}s^{-1}$ within the surface pressure minimum shows that definite cyclogenetic forcing is induced by the PV anomaly situated closer to the surface.

The results of Experiment 2 clearly show that the effective height of the intrusion AGL is a massive factor in the amount of surface cyclogenetic forcing that is induced. With the same intrusion vertical depth, the experiments with lower intrusion

height AGL result in more intense lowering of the surface pressure. It should be noted that these tests were also repeated with different vertical depth of intrusions by using different anomaly radial heights of 2500m and 7500m (not shown). The tests show a similar result where intrusions associated with the higher tropopause result in a lesser amount of cyclogenetic forcing than those associated with the lower tropopause.






**Figure 10: Same as in Figure 8 with the exception that in this case the anomaly radial height is kept constant at 5000m with varying tropopause heights of 15000m (row 1), 12500m (row 2), as in Experiment 0) and 10000m (row 3).**



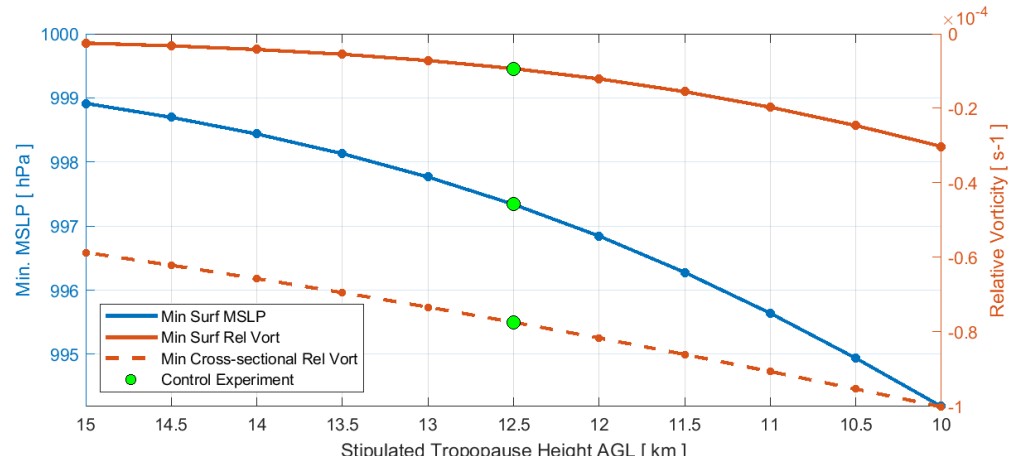

**Figure 11: Similar to Figure 9 but for Experiment 2 (varying tropopause height experiments) with tropopause height AGL shown on the x-axis**

### 3.4 Experiment 3: Constant intrusion height from varying tropopause height and intrusion depth

The results in Experiments 1 and 2 show that the proximity of stratospheric intrusion to the surface has a larger impact on surface low deepening and surface cyclogenesis than the vertical intensity or size of the intrusion itself. This was also hypothesized by Barnes et al. (2021a) with respect to COL vertical extensions. In order to confirm this concept, anomalies are created such that they extend to a similar height AGL from a varying tropopause height and compared to Experiment 0 (Figure 12-A2 and B2). In this case anomalies with radial heights of 7000m, 5000m and 3500m were introduced into the fields with tropopause heights of 15000m, 12500m and 10000m AGL respectively. The resulting intrusions heights AGL were calculated to be at 6750m, 6594m and 6350m AGL respectively. All three of these intrusions induce cyclonic motion around the anomaly of similar intensity, with maximum meridional wind velocities between 11-12m.s⁻¹. The intrusions also all induce a similar surface level pressure deepening resulting in about a 3hPa surface central pressure drop.

Some small, but noticeable differences can however be seen between the different scenarios in Experiment 3. The larger intrusion emanating from a higher dynamical tropopause (Figure 12-A1), results in a slightly deeper penetration of the high-velocity core around the simulated anomaly compared to lower intensity anomalies in Figure 12-A2 and A3. In Figure 12-A1 the 4m.s⁻¹ contour extends to a depth 1000m deeper than in Figure 12-A2 and A3. The effect of this is also noticeable on the surface. Relative vorticities on the surface increase with intrusion vertical intensity from sub-cyclogenesis values in Figure 12-A3 to weak cyclogenesis values in Figure 12-A1. A major difference between these scenarios is the presence of a -2PVU anomaly within the intrusion in Figure 12-A1 that does not appear in Figure 12-A2 or A3. This is an artefact of the basic state setup but could be influencing and enhancing the additional rotation at the surface. The influence of anomaly amplitude will be further investigated in Experiment 4.

**Figure 12: Same as in Figure 8 but with variable anomaly radial heights such that the height of the stratospheric intrusions AGL are similar from varying tropopause heights of 15000m (row 1) and 10000m (row 3). Experiment 0 is given in row 2 for ease of reference.**

### 3.5 Experiment 4: Varying anomaly magnitude

Experiment 3 brings forth the question of the magnitude intensity of the stratospheric intrusion with respect to its effect on the
cyclonic circulation at all tropospheric levels around the anomaly. Experiment 4 tests this effect by changing the magnitude of
the intrusion, ie. by varying the anomaly amplitude. A lower (-1.0 PVU) and higher (-2.0 PVU) scenario are tested and shown
in Figure 13. For ease of reference, the -1.0 PVU contour is also plotted as a dashed magenta line in Figure 13. As our definition
of the tropopause continues to be -1.5PVU, the resulting intrusion of the lower scenario is very shallow, but does contain a
small anomaly close to the depth of Experiment 0 (Figure 13-A2). Figure 14 shows that the magnitude of the intrusion has
some effect on the mid-tropospheric cyclogenetic forcing. Minimum cross-sectional relative vorticity decreases by $3\times10^{-5}s^{-1}$
from the low to high anomaly amplitude scenarios, whilst the maximum meridional velocity decreases by $1m.s^{-1}$ around the



anomaly. The weaker circulation does result in a slightly lower relative vorticity at the surface (less by only $4\times10^{-7}s^{-1}$). Comparable surface pressure lowering still results beneath the anomaly. For the higher magnitude scenario, the opposite is true. A slight strengthening (by similar magnitudes) are seen in both the circulations around the anomaly (increase in maximum

meridional velocity of $1m.s^{-1}$) and at the surface (increase in the relative vorticity by $3\times10^{-7}s^{-1}$). The results of Experiment 4 reaffirm the findings in experiment 3, ie. that the vertical intensity of the stratospheric intrusion could play a more vital role in affecting surface circulation than the magnitude of the PV intrusion.



**Figure 13: Same as in Experiment 0 with varying anomaly magnitudes of -1.0 PVU (row 1) and -2.0 PVU (row 3). Experiment 0 (with an anomaly magnitude of -1.5 PVU) is shown in row 2. In addition to the -1.5 PVU contour (thick magenta line), the -1.0 PVU contour is also provided for context by a dashed magenta line.**



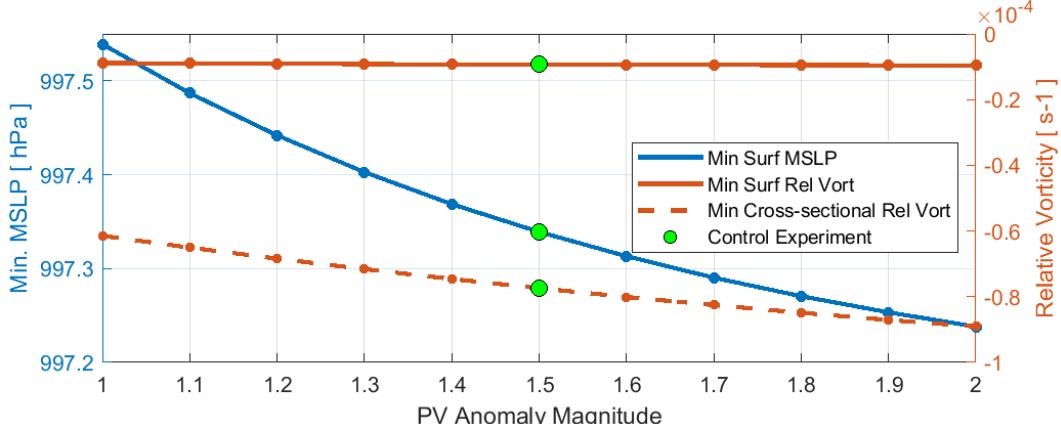

**Figure 14: Same as in Figure 9 but for Experiment 4 (varying stratospheric intrusion magnitude experiments)**

## 3.6 Experiment 5: Varying anomaly horizontal width

RWB events, which drive stratospheric air into the troposphere, have been classified into 4 distinct categories, namely cyclonic equatorward breaking (LC2), cyclonic poleward breaking (P1), anticyclonic equatorward breaking (LC1) and anticyclonic poleward breaking (P2) (Thorncroft et al. 1993; Peters and Waugh 1996). These different types of RWB are associated with differing characteristics of the isentropic PV filaments produced and therefore differences in the geometric characteristics of the PV intrusions and the weather patterns they produce. Elongated, thin filaments of stratospheric air are produced by LC1 type breaking events because of the anticyclonic shear on the equator side of the PV structure (Thorncroft et al. 1993). These associated PV filaments eventually roll up and often produce cut-off PV structures. P1 events are also associated with thinner filaments of high-PV air (Peters and Waugh 1996). Similarly to P2 events respectively, LC2 events are associated with broader high-PV streamers which, under the influence of cyclonic shear, wrapping cyclonically and not resulting in cut-offs (Thorncroft et al. 1993). Recent examples of the different breadth of these streamers can be seen in zwhere a thin PV streamer (of ~$1^o \approx 100km$) and broader streamer (of ~$10^o \approx 800km$) affected and deepened a surface cyclone.

The effect that the width of the intrusion has on surface cyclogenesis is tested with a numerical setup similar to that of Figure 4 with varying anomaly radial widths. Selected width experiments are shown in Figure 15 and a summary of the effect of all experiments effect on surface parameters is shown in Figure 16. All stratospheric intrusions were defined such that they reach a similar depth (using a constant 5000m anomaly radial height from the 12500m AGL dynamical tropopause).

Clear differences in the circulation around the anomaly can be seen in Figure 15A and Figure 16. The thinner intrusion (100km width) in Figure 15-A1 results in a much weaker circulation than the standard configuration (Experiment 0, Figure 15-A2) with maximum meridional velocities of 8m.s$^{-1}$ (compared to 11m.s$^{-1}$ in Experiment 0). Conversely, stronger cyclonic circulation is induced around the broader anomaly with a maximum meridional velocity of 15m.s$^{-1}$. The core of the cyclonic circulation is also augmented by the breadth of the intrusion. The thinner intrusion results in a thinner and shorter jet core around the anomaly, whilst the broader anomaly extends the core both vertically and horizontally, affecting almost the entire





cross-sectional domain. The enhanced circulation can also be seen in the minimum cross-sectional relative vorticity associated with different intrusions in Figure 16. The minimum cross-sectional relative vorticity data clearly shows a decrease in the intensity of the mid-level cyclogenetic forcing around the anomaly. The increase in the minimum cross-sectional relative

vorticity with increasing width of the anomaly is exponentially increasing from over $1 \times 10^{-4} s^{-2}$ to $3 \times 10^{-5} s^{-2}$.

The width of the intrusion is also important to surface parameters. The thinner intrusion results in a reduction of cyclogenetic forcing with no discernible closed circulation, a much lower central surface pressure (1hPa) and a decrease in the magnitude of relative vorticity (Figure 15-B1). The broader intrusion however enhances cyclogenetic forcing at the surface compared to Experiment 0. Doubling the breadth of the intrusions results in a much deeper surface low pressure (8hPa deeper than

465 Experiment 0). Figure 16 shows that the surface pressure minimum decreases quasi-linearly with increasing PV anomaly width. The differences in the relative vorticity on the surface are however much less significant. Only a slight decrease in the minimum relative vorticity on the surface is discernable between the thin and broad scenarios (Figure 16).

**Figure 15: Same as in Figure 8 but with variable anomaly radial widths such that the height of the stratospheric intrusions AGL are similar from a constant dynamical tropopause depth of 12500m. The thinner intrusion is created by an anomaly with a 100km radial width (row 1) whilst the broader intrusion is created by a 400km radial width (row 3). Experiment 0 (200km radial width) is provided in row 2 for convenience and comparison.**

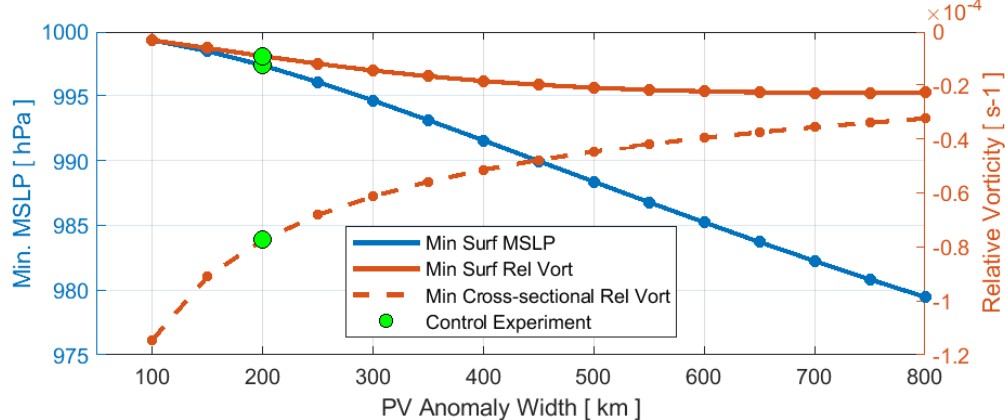

**Figure 16: Same as in Figure 9 but for Experiment 5 (varying anomaly radial width experiments)**

## 4 Discussion and conclusion

The study by Barnes et al. (2021a) showed a climatological link between COL depth and stratospheric intrusion depth. Deep COLs, those associated with surface cyclones, were generally found to be associated with deeper stratospheric intrusions. The climatology Barnes et al. (2021a) is performed in the context of the total atmospheric system, where a host of processes at various levels of the atmosphere can affect the development of cyclones at all levels. Therefore, it is pertinent to isolate the link between COLs and the depth of their associated upper-level high-PV anomalies with surface cyclogenetic forcing in an idealised setting. An initial numerical experiment using PV inversion diagnostics (Experiment 0) shows that the numerical model and PV inversion algorithms produce results as we expect from the conceptual model (Figure 1) of a high-PV anomaly in the upper-levels: a core of cyclonic flow is present around the PV anomaly, with general cyclonic motion prevalent, although weak, through the majority of the domain. The longitudinal cross-section and upper-level pressure and wind fields show the development of an amplifying trough. In the real world, the trough with continual amplification could develop into a COL. Cyclonic flow and low-pressure signatures are also observed on the surface. This re-emphasises that upper-level processes are a shared development driver of both the surface cyclone and its associated COL. For Experiment 0, cyclogenetic forcing is very weak or negligible, although relative vorticities indicate that we are on the precipice of surface cyclogenetic forcing being induced. The anomaly does however induce a closed low-pressure on the surface.

Experiment 1 varies the depth of a stratospheric intrusion (ie. the closest distance of a -1.5 PVU point to the surface). This experiment reveals that the depth that a stratospheric intrusion reaches is an important factor in the amount of surface cyclogenetic forcing induced. Very shallow intrusions resulted in a minimal amount of pressure decrease on the surface, whilst extremely deep intrusions resulted in a pronounced decrease in surface pressure. Surface relative vorticities reveal that surface cyclogenetic forcing is only initiated for deeper intrusions. Barnes et al. (2021a) showed that COLs are more likely to be associated with a surface low if a stratospheric intrusion reaches below the 300hPa level. Using a simple barometric conversion from altitude to pressure with standard sea level pressure and temperature reveals that the stratospheric intrusions shown in



Experiment 1 (Figure 8) extend to roughly the 270hPa, 430hPa, 610hPa and 840hPa levels. This corresponds well with the findings of Barnes et al. (2021a) with the only intrusion extending to less than the 300hPa, having little effect on the surface. The lower limit for weak cyclogenetic forcing could be deemed to be the control experiment since it is extremely close to the lower limit of cyclogenesis. Therefore, in Experiment 1 cyclogenetic forcing is initiated just past the 430hPa intrusion level. This relative vorticity view of cyclogenetic forcing therefore also corresponds well to the findings of Barnes et al. (2021b).

The scenarios in Experiment 1 show how surface cyclogenetic forcing is induced for separate instantaneous intrusions. However, Figure 8, can also be viewed in a temporal sense. An RWB event, results in an intrusion of high-PV air into the stratosphere. The continued amplification of the Rossby wave will result in a continued intrusion of high-PV air into the troposphere. The intrusion will grow and extend towards the surface as shown in Figure 8A. The growth in the intrusion will result in the continued and enhanced development of the surface cyclone and if deep enough, could result in the cyclogenesis. Barnes et al. (2021a) showed that shallow COLs, COLs which only extend into the mid-levels, occur most frequently in the summer months and the lower latitudes. This corresponds to seasons and regions where the dynamical tropopause is furthest away from the surface. This finding together with the finding that shallow COLs are most often associated with shallow intrusions suggest that the height of the stratospheric intrusion is more important than the vertical depth of the intrusion itself. Experiments 2 and 3 show this is indeed the case. Intrusions from high tropopause heights, as would be seen closer to the tropics and in summer, resulted in negligible cyclogenetic forcing at the surface and initiated very little pressure decrease at the surface (Experiment 2). Conversely, lower dynamical tropopauses resulted in surface cyclogenetic forcing and a large surface pressure decrease. Differing intrusion depths to a similar intrusion height AGL were also shown to result in similar pressure deepening (Experiment 3). It was however shown the cyclonic motion at the surface was more enhanced for the larger vertical intrusions compared to the smaller vertical intrusion. The enhanced relative vorticity in the large intrusion suggests that the vertical height of intrusion could play a role in the extreme windstorms (eg. Liberato 2014). Of course, it should also be noted that anomalies at the surface and in the low-levels can also enhance cyclogenesis when in phase as shown in the example of Cape Storm by Barnes et al. (2021b).

A key finding of this study is the relative contribution of each of the different factors tested to surface cyclogenesis. Experiments 1-3 show that it is the depth that the stratospheric intrusion reaches that is the major factor in surface cyclogenesis. Larger intrusions induce greater cyclogenetic forcing in the mid-levels than smaller intrusions. However, if the PV intrusions are situated further away from the surface (from a tropopause further away from the surface), the resulting relative vorticity on the surface is diminished and is comparable to that of a smaller intrusion extending to a similar height AGL from a tropopause height closer to the surface. In terms of surface relative vorticity, wider intrusions are found to have a small, but largely negligible effect, despite the dramatic effect that these differences have in the mid-troposphere. The width of these intrusions do however still have a large effect the surface pressure of beneath them. With constant stratospheric depth (5000m) and constant tropopause height AGL (12500m), a thin filament intrusion with a radius of 100km resulted in a minimal pressure decrease at the surface. Interestingly, the degree of surface pressure was comparable to that of the intrusion of double the width (200km) and half the vertical extent (2500m) as seen in Experiment 1 (Figure 8, top panels).



Conversely, an intrusion with a large area of with radius of 400km resulted in a slightly deeper surface low-pressure comparable to that of the intrusion with half the width and 50% more depth as shown in Experiment 1 (Figure 8, bottom panels). The resulting surface pressure and relative vorticity patterns together provide a picture of the weather systems being forced on the surface. Wider intrusions produce wider, deeper cyclones whilst thinner filaments of PV produce shallower,
smaller cyclones.

Conceptually, there is a distinct difference between the surface cyclonic circulations induced by the idealised intrusions presented here and which actually occur in reality. This is explained fully in Hoskins et al. (1985) and was touched on in Barnes et al. (2021a). Surface lows are generally found to the east of the COL and upper-level PV anomaly axis. In these idealised cases however, the centre of the induced surface cyclonic is directly beneath the PV anomaly. In the real atmosphere,
the surface cyclonic motion induced by the upper-level anomaly acts as a mechanism for warm surface temperature advection to the east of the upper-level anomaly. This surface temperature (and therefore potential temperature) anomaly has PV-like properties and can induce its own surface cyclonic circulation. Deeper intrusions will therefore drive more intense warm-air advection to the east of the trough axis, inducing enhanced cyclogenesis. One of the major limitations of this work is the lack of a temporal aspect in the experimental framework. Surface cyclones are not produced instantaneously but grow over time.
Additionally, in the real-world atmosphere, upper-level PV anomalies are also influenced by the vertical structure of the air column beneath it. Future work should include the use of a numerical dynamical core which will have a temporal element and include processes such as upper-level induced surface warm-air temperature advection. This would also improve general analysis of the temporal aspect of intrusions as they grow and decay and the resulting effect on surface cyclogenesis. A dynamical core would also allow for the study of more complex vertical structures such as the inclusion of temperature
inversions beneath the PV inversion.

This study analyses idealised stratospheric intrusions in a general sense. Although the climatology of Barnes et al. (2021a), looked at PV anomalies with respect to COL development, they are applicable for both intrusions that result in the formation of a closed low as well as those that only ever form an upper-level trough. Regardless, the results clearly confirm the climatological link found by Barnes et al. (2021a) that stratospheric intrusion depth is an important contributing factor that
results in the extension of a COL to the surface.

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
