# Peer review of "Stratospheric intrusion depth and its effect on surface cyclogenesis: An idealized PV inversion experiment"

_Weather and Climate Dynamics, 2021_

## Referee Comment (RC1)

**Review of WCD-2021-24: Stratospheric intrusion depth and its effect of surface cyclogenesis: An idealize PV inversion experiment**

**Overview:** This manuscript sought to explore the role of Southern Hemisphere stratospheric intrusions (lowering of the stratosphere into the troposphere, realized as a lowering of the dynamic tropopause) on tropospheric circulations. The authors used a simplified atmospheric state to compute a variety of PV inversion experiments examining the relative impacts of a deeper intrusion, an intrusion from different tropopause heights, a combination of the two, a stronger intrusion, and a wider intrusion. The results are generally as expected from existing PV thinking – the closer the cyclonic PV anomaly to the surface, the stronger the influence at the surface. Some interesting results were either affirmed or introduced (in particular regarding the intrusions from a higher dynamic tropopause, and the role of wider/narrower PV anomalies), but in general the results re-affirmed known relationships that are applicable to either hemisphere. I hesitate to say that the article isn't introducing anything new – often these relationships are taken for granted through theoretical arguments but aren't shown through simple modelling exercises – but little about the results seemed new. Perhaps more importantly, the connections drawn by the authors to both cyclogenesis (rather than just a cyclonic response to a PV anomaly) as well as to real-world scenarios were not robust enough to make it clear that this presented a substantial contribution to the science. There were also concerns about the model set-up in the first place, which appeared simplified to the point of violating key laws of the mid-latitude dynamics that appeared to be the focus of the manuscript. Lastly, the manuscript was written too colloquially and lacked focus in many places. I do believe that with a careful and attentive revision the paper will represent a good contribution to the field, and as such I recommend a major revision and re-review.

**Major Comments:**

1. *Model and Experiment Set-up:* The study seeks to study, and compare to, mid-latitude dynamic processes (a prescribed latitude of 43°S) that occur in a baroclinic environment, but the model itself is run as a barotropic model. The authors need to clearly justify their model set-up and how to interpret their results appropriately. In the Barnes et al. 2021a paper that they base the climatology off of, the authors made clear that the COLs of interest occur in a baroclinic environment (which is required for the jet to occur under thermal wind balance). The reviewer recognizes that idealized barotropic models are an important tool in trying to diagnose these questions, but the authors need to make a much more clear and stronger justification for doing so. Regarding the experiment set-up, the authors also to more clearly justify their decisions for the spatial extent and intensity of the PV anomalies. There are many studies out there that have studied PV anomalies, so a more clear justification for why they determined a PV anomaly to 'look' the way it did is necessary. Of particular focus needs to be a clear justification of the horizontal and vertical extent, as well as a justification for a sudden and total relaxation of the PV gradient in the anomaly itself once the -1.5 PVU threshold is met (eg. Fig. 4).

2. *Conflation of the idea of cyclogenesis and a cyclonic response:* The authors routinely refer to a cyclogenesis term (based on thresholds of cyclonic vorticity – lines 213-215) but are looking at the response at the surface given the existence of a PV anomaly at the tropopause. In other words, by using a PV inversion (rather than integrating a model forward where a PV anomaly is introduced, and the surface is allowed to evolve in response), you are not looking at

cyclogenesis, but instead the existence of a cyclonic circulation due to the existence of a PV anomaly. This undermines several components of the results, both when the authors discuss the surface circulation due to the PV anomaly as well as when they make points on the potential evolution of the anomaly were it allowed to evolve in time. Further, their discussion of these points leave concern about a lack an understanding of PV dynamics. If the PV anomaly and surface cyclone are vertically stacked (as they are in the results), and this is a dry barotropic environment, the vorticity anomalies are the only factor at hand that can influence the system, meaning only movement is allowed rather than intensification. Thus, the surface cyclone cannot undergo further development from an intensification standpoint, and the same holds for the upper level cyclone – thus leaving the question of how the system can ever 'develop its own closed, cyclonic circulation (or COL)' (line 303). A very careful examination and reworking of this discussion is critical for the interpretation of the results.

3. *Dynamic interpretation/explanation of experimental results:* The results here are interesting – but lack a fair bit of interpretation from a PV framework regarding why the responses are occurring. The authors make some efforts on this front, but more needs to be done beyond just reporting the results to really enhance the impact of this study. For example, experiment 4 (changing the intensity of the PV anomaly) shows almost no change in response despite a presumably stronger PV gradient (though it might not be that much stronger given the experiment set-up). The results are interesting – but there's little to no interpretation for why we see the response we do. The same goes for experiment 5 – the authors report the change in tropospheric circulation but provide little interpretation for why. For example – why do we see a decrease in cyclonic relative vorticity with a widening PV anomaly? How can this be interpreted in a PV framework? Why is the surface circulation so much stronger?

**Minor comments:**
*General comments:*
- The manuscript reads *very* colloquially which is problematic. Please carefully check through the manuscript to identify instances where this occurs – I've identified some examples here, but there are many others throughout:
    - L40: Air can be advected or diabatically altered, but it cannot be 'introduced' to another region of the atmosphere.
    - L42-43: The term 'basic' here isn't necessary, and acts to undermine your study (there's little basic about PV theory – it's an advanced synoptic-dynamic topic that readers may not be familiar with).
    - L67-80: You use the term 'This study examines/looks at/aims' too much here – aim to rework a bit.
    - L92: 'diagnostic for reanalysis sets to diagnose' – aim to avoid repetitive words in a single sentence (there were several of these in the manuscript)
    - L237-238 and elsewhere: Unless quantifying, avoid using the term 'stronger' and 'weaker' or similar qualitive statements (other examples include 'meagre' in L370 or 'massive' in L378)

- o L291 and elsewhere: The term 'exponentially' refers to a mathematically derived curved for a set of data points – if it is exponential, prove it; otherwise, please avoid statements that imply something different from what the data shows.
  - o L362 and elsewhere: Aim to avoid injecting opinion or emotion – lines such as 'Since we are dealing with …' should be avoided.
- Definition of terms: In several instances, terms/acronyms were introduced but not defined. As a reader familiar with the topic, I could ascertain nearly everything, but less familiar readers may struggle. Examples include (but aren't limited to):
  - o PVU (L37)
  - o COL (L64)
  - o Reference state (L86) – be sure to define what this is and how you establish it
  - o "halo" (L150)
  - o MSLP (L201)
  - o Sphere of influence (L271)
  - o Mid-tropospheric (L347)
  - o Total atmospheric system (L478)
- Figures/equations: There were several inconsistencies in the figures that could be tightened up, and equation 6 does not need to be there (it's just a re-arrangement of equation 1 and can be stated as such):
  - o Figure 3 and elsewhere: Please always use panel labels (eg. A and B) consistently. Please also include reference vectors whenever showing vectors that represent direction and magnitude.
  - o Figure 5: This would be more clearly communicated as a table rather than a flow chart
  - o Figures 9, 11, 14, and 16: Please use the same axis labels amongst these four figures. This is particularly important for Figure 14, which appears to have a large MSLP response based on figures 9, 11, and 16, but in reality is only ~0.3 hPa.
  - o All captions: Be sure to include all relevant information, such as MSLP contour intervals or wind speed contour intervals.

---

## Referee Comment (RC2)

**"Stratospheric intrusion depth and its effect on surface cyclogenesis: An idealized PV inversion experiment"**

**Authors:** Barnes, Ndarana, Sprenger, and Landman

**Recommendation:** Major Revision

**Overview:**

In this study, the authors perform a series of idealized experiments in which they invert QGPV anomalies of various sizes, shapes, and vertical depths for their associated horizontal circulations. These circulations are then used to identify QGPV configurations that are likely to be more influential on surface cyclogenesis. In my opinion, while the analysis does not necessarily offer any new qualitative dynamical insight beyond what has already been garnered from the application of a PV framework in prior work, the experiments performed herein do provide a nice systematic quantitative treatment of how nuances in the structure of QGPV anomalies contribute to surface cyclogenesis. This quantitative assessment is novel from my perspective, and justifies the value of this study.

That being said, there several instances within the text in which I felt the present work could be better motivated and described with improved precision. Additionally, the intensification rate of surface cyclones is an important component of the interaction between upper-level PV anomalies and the surface. The temporal evolution of surface cyclones is not considered as part of the analysis but is quantifiable using diagnostic PV tendency inversions. Last, the analysis does not necessarily consider the role that static stability plays in modulating the character of these interactions and may represent an additional experiment that the authors can consider integrating into their analyses. Given the extent of my comments below, I have recommended the manuscript be returned for Major Revisions.

**Major Comments:**

**1.** There were several instances within the text in which I found myself a bit confused regarding the interpretation of figures (see minor comments below). I believe that this confusion could be remedied with a thorough review of the text to improve the precision of the discussion and through better definition of various terms. For example, it was difficult for me to differentiate between the physical interpretation of the minimum relative vorticity and the cross-sectional minimum in relative vorticity.

**2.** The introduction and motivation for the present work could be made clearer. Namely, it might be effective to construct a figure that highlights the diversity of PV intrusions and how these structures are associated with different surface cyclone intensities in real data. This figure could more effectively frame the idealized experiments performed in this study. There are also several instances in the introduction where the authors emphasize that such a study has not been performed in the Southern Hemisphere. But, to my knowledge, there is no reason to expect that PV anomalies will behave in a dynamically different way compared to the

Northern Hemisphere. Therefore, I recommend this discussion should be either more strongly motivated or eliminated from the text.

**3.** The authors do not perform a temporal diagnosis of the evolution/development of cyclonic circulations at the surface in their idealized experiments, but such an analysis can be performed in a diagnostic sense using either QGPV (e.g., Breeden and Martin 2018) or Ertel PV (e.g., Davis and Emanuel 1991). I wonder whether the application of this diagnostic framework for examining the instantaneous intensification rate of surface cyclones would bolster the analysis. It is also not clear in the text why QGPV is adopted over Ertel PV. At the very least, this choice should be justified in the context of the proposed applications.

**4.** One element that is not explicitly considered in the idealized experiments is the role of static stability. Namely, those PV anomalies that are situated in a less stable environment are able to more effectively induce cyclogenesis. Could an experiment be run that considers varying the static stability of the environment? Additionally, the inversion of QGPV requires the specification of a reference atmosphere. It is not clear from the text what the authors have selected as their reference atmosphere, unless I may have missed it.

**5.** It is somewhat difficult to compare the various experiments because the summary figures (e.g., Fig. 9) feature different values along their y-axes. To better enable a comparison between experiments, I'd recommend standardizing these y-axes across all similar plots.

**Minor, Specific, and Typographical Comments:**

*Abstract*
L15–16: I am a bit confused by the discussion in these lines. Namely, L15 states that horizontal extent is more important, whereas the next line states that vertical depth is important in dictating the strength of the circulation – both of which can be used to characterize cyclone intensity. Could these lines be clarified to better describe the respective influences of the vertical and horizontal extent of the stratospheric PV anomalies?

L17–18: This relationship in this sentence has to be true by definition, and I wonder whether it could be deleted to make more room to better clarify the nature of the relationships described in L15–16.

*1. Introduction*
L41: The specification that "high PV" corresponds to negative values should occur earlier in the manuscript when this terminology is first used. I also believe it could be made clearer earlier in the introduction or abstract that the focus of the manuscript will be on Southern Hemispheric anomalies to avoid any potential confusion imparted on a reader.

L44–45: The vertical depth of the circulation induced by the PV anomaly in Fig. 1 is also a function of the static stability. Namely, the lower static stability in the troposphere compared to the stratosphere allows the circulation induced by the PV anomaly to penetrate deeper towards the surface. Some reference to the thermodynamic environment in which the PV anomaly is embedded could benefit the discussion at this juncture in the text.

L51:  In the context of this study, cyclogenesis is described as a near-surface phenomenon. Consequently, I found the reference to cyclogenesis occurring throughout the stratosphere to be a bit confusing. Could this line be revised for improved clarity?

L61: Why is it expected that the influence of PV anomalies will be different in the Southern Hemisphere compared to the Northern Hemisphere? I believe this claim may require stronger motivation/explanation.

L62–63: The term "stratospheric tropopause" is not accurate, since the tropopause represents the interface between the troposphere and the stratosphere.

L64: The acronym, "COL", has not yet been defined in the manuscript.

L64–66: It is not clear to me how this conclusion follows from the previous sentences in the paragraph. Consider a revision to improve the clarity of the discussion.

L72: As in L61, it is not clear why the physical influence of cyclonic PV anomalies will differ between the Northern and Southern Hemisphere simply because the sign of PV is negative in the Southern Hemisphere.

L72–73: This sentence is somewhat redundant with the statement that ends in L70. Consider whether it could be deleted.

*2. Methodology*
L89: The Davis (1992) study focuses on the inversion of Ertel PV rather than QGPV, for which the system of equations for performing the inversion features nonlinear terms. For QGPV, the differential operator is linear, which does return a unique solution using successive over-relaxation.

L114–119: I found this discussion to be a bit confusing. Could you clarify more as to why the tropopause is defined differently within the inversion algorithm?

L121: Is the "specified DT" the height of the tropopause above ground level needed for the algorithm or the −1.5 PVU isosurface?

L127: Is the intent to refer to the left panel of Fig. 3 in conjunction with this discussion? I ask because the right panel of Fig. 3 does not show any pressure contours.

L205–210: I found this discussion concerning vorticity thresholds to be a bit confusing. Are you basically looking for areas near the surface that feature vorticity with the magnitude described in the text? Or are you looking for areas where the circulation induced by the upper-level PV anomalies features vorticity of a particular magnitude at the surface.

L213–215: To what extent are the results sensitive to the selection of these vorticity thresholds?

*3. Results*
L244: If referring to a line of constant pressure here and elsewhere in the manuscript, "isobar" is more descriptive than "isohypse".

L297–298: This particular sentence, as currently written, is a bit confusing. Would it be possible to rewrite it for further clarity? Similarly, I found L300–301 to also be confusing, which may require an edit for further clarity.

L359–365: Could a plot of static stability be produced to help illustrate this effect more clearly?

L392: Arguably, this claim may be best reserved until after the final few experiments have been introduced (i.e., anomaly intensity has not been considered yet). Unless the goal here is to refer to the vertical extent of the anomaly. If so, a revision may be necessary to make that point clearer.

L406: Would it be possible to expand further on how this result may be an artifact of the basic state?

L421–427: I'm having a bit of a difficult time verifying some of these values against those plotted in Fig. 14. Could the text be revised to more clearly reference where these results are drawn from.

L436–440: Could a figure be produced that shows the characteristic PV structure associated with these categories in real cases. This may help to visually motivate the forthcoming experiment.

L445–446: It is not clear what this particular statement is referring to.

L459–460: This statement is a bit difficult to verify. Namely, Fig. 16 suggests that the magnitude of the relative vorticity decreases with increasing width of the anomaly, which is counter to the discussion in this section. I think my confusion here may stem from difficulty understanding how the cross-sectional relative vorticity is calculated/defined.

L462: For greater specificity, it may be worth stating that this 1 hPa corresponds to a pressure *perturbation* – as a "lower central pressure" would typically correspond to a stronger storm.

*4. Discussion and Conclusion*
L520–525: These few lines may be a bit redundant with the discussion in the previous paragraph. Consider whether these lines could be deleted without any loss of content.

*Figures and Tables:*
Fig. 2: Would it be possible to specify the contour interval for PV in the caption?

Fig. 5: It is not clear to me from the flow chart why all the experiments connect with the basic state box. It may be clearer to put the basic state box at the top of the image and then have all experiments flow beneath it.

Fig. 6: Could the contour interval for the meridional wind be included as part of the caption?

Fig. 9: It is not clear to me how the cross-sectional minimum relative vorticity is different from the minimum relative vorticity value. Could this be explained a bit more clearly in the text?

Fig. 9/11/13: Would it be possible to make the limits along the y-axes the same in all of these plots so as to allow for better comparison between experiments?

References:

Breeden, M., and J. E. Martin, 2018: Analysis of the initiation of an extreme North Pacific jet retraction using piecewise tendency diagnosis. *Quart. J. Roy. Meteor. Soc.*, **144**, 1895–1913, https://doi.org/10.1002/qj.3388.

Davis, C. A., and K. A. Emanuel, 1991: Potential vorticity diagnostics of cyclogenesis. *Mon. Wea. Rev.*, **119**, 1929–1953, https://doi.org/10.1175/1520-0493(1991)119<1929:PVDOC>2.0.CO;2.

---

## Author Comment (AC1)

**Review of WCD-2021-24: Stratospheric intrusion depth and its effect of surface cyclogenesis: An idealize PV inversion experiment**

**Overview:** This manuscript sought to explore the role of Southern Hemisphere stratospheric intrusions (lowering of the stratosphere into the troposphere, realized as a lowering of the dynamic tropopause) on tropospheric circulations. The authors used a simplified atmospheric state to compute a variety of PV inversion experiments examining the relative impacts of a deeper intrusion, an intrusion from different tropopause heights, a combination of the two, a stronger intrusion, and a wider intrusion. The results are generally as expected from existing PV thinking – the closer the cyclonic PV anomaly to the surface, the stronger the influence at the surface. Some interesting results were either affirmed or introduced (in particular regarding the intrusions from a higher dynamic tropopause, and the role of wider/narrower PV anomalies), but in general the results re-affirmed known relationships that are applicable to either hemisphere. I hesitate to say that the article isn't introducing anything new – often these relationships are taken for granted through theoretical arguments but aren't shown through simple modelling exercises – but little about the results seemed new. Perhaps more importantly, the connections drawn by the authors to both cyclogenesis (rather than just a cyclonic response to a PV anomaly) as well as to real-world scenarios were not robust enough to make it clear that this presented a substantial contribution to the science. There were also concerns about the model set-up in the first place, which appeared simplified to the point of violating key laws of the mid-latitude dynamics that appeared to be the focus of the manuscript. Lastly, the manuscript was written too colloquially and lacked focus in many places. I do believe that with a careful and attentive revision the paper will represent a good contribution to the field, and as such I recommend a major revision and re-review.

**Major Comments:**

1. *Model and Experiment Set-up:* The study seeks to study, and compare to, mid-latitude dynamic processes (a prescribed latitude of 43°S) that occur in a baroclinic environment, but the model itself is run as a barotropic model. The authors need to clearly justify their model set-up and how to interpret their results appropriately. In the Barnes et al. 2021a paper that they base the climatology off of, the authors made clear that the COLs of interest occur in a baroclinic environment (which is required for the jet to occur under thermal wind balance). The reviewer recognizes that idealized barotropic models are an important tool in trying to diagnose these questions, but the authors need to make a much more clear and stronger justification for doing so.

We thank he reviewer for this suggestion of adding in more justification for our use of experimental design. The idea behind this choice of barotropic model was made in order to strip the environment of any other forcings and isolate the PV intrusion itself. We would like to test PV intrusions in a baroclinic environment as well but it is felt that this could be for further study which include temporal aspects and additional advection aspects as well. We have added this aspect into the manuscript

Regarding the experiment set-up, the authors also to more clearly justify their decisions for the spatial extent and intensity of the PV anomalies. There are many studies out there that have studied PV anomalies, so a more clear justification for why they determined a PV anomaly to 'look' the way it did is necessary.

Of particular focus needs to be a clear justification of the horizontal and vertical extent, as well as a justification for a sudden and total relaxation of the PV gradient in the anomaly itself once the -1.5 PVU threshold is met (eg. Fig. 4).

The relaxation of the PV gradient within the anomaly itself was chosen for ease of controlling the magnitude of the PV as shown in the resulting PV intrusion. In these experiments we use the

PV anomaly to attempt to recreate PV intrusions as closely as possible to as would be seen in reality ie. A tongue of high magnitude PV extending from the stratospheric pool of high PV. Within the numerical framework used. Restricting the gradient within the anomaly was the most effective way at ensuring a tongue of -1.5 PVU resulted. We do agree that there would be a PV gradient in a real-world PV intrusion. In Experiment 4, the magnitude test looks to test this out by adding additional PV values within the 1.5 PVU and shows that it results in little effect on the cyclogenetic forcing induced. We have added this argument into the manuscript during our discussion of the anomaly in the experimental setup.
"It is acknowledged that within real-world PV intrusions, there would be a PV gradient within the PV intrusion lower than PV intrusion boundary is defined. However, within the experimental framework the interior of the PV intrusion is kept constant to more easily control the magnitude of the PV intrusion that results from the PV anomaly."

2. *Conflation of the idea of cyclogenesis and a cyclonic response:* The authors routinely refer to a cyclogenesis term (based on thresholds of cyclonic vorticity – lines 213-215) but are looking at the response at the surface given the existence of a PV anomaly at the tropopause. In other words, by using a PV inversion (rather than integrating a model forward where a PV anomaly is introduced, and the surface is allowed to evolve in response), you are not looking at cyclogenesis, but instead the existence of a cyclonic circulation due to the existence of a PV anomaly. This undermines several components of the results, both when the authors discuss the surface circulation due to the PV anomaly as well as when they make points on the potential evolution of the anomaly were it allowed to evolve in time. Further, their discussion of these points leave concern about a lack an understanding of PV dynamics. If the PV anomaly and surface cyclone are vertically stacked (as they are in the results), and this is a dry barotropic environment, the vorticity anomalies are the only factor at hand that can influence the system, meaning only movement is allowed rather than intensification. Thus, the surface cyclone cannot undergo further development from an intensification standpoint, and the same holds for the upper level cyclone – thus leaving the question of how the system can ever 'develop its own closed, cyclonic circulation (or COL)' (line 303). A very careful examination and reworking of this discussion is critical for the interpretation of the results.

The authors understand the confusion of our arguments with respect to cyclogenesis and cyclogenetic forcing. We have removed our interpretation of the cyclogenetic forcing resulting at the surface with respect to cyclogenesis to remove the temporal implications of our arguments. Instead with only compare the cyclogenetic forcing in terms of pressure reduction and cyclonic vorticity that results on the surface due to the existence of the PV anomaly comparatively between the experiments. We do acknowledge that future work should look at the temporal aspect of these PV intrusions utilizing a dynamical core or similar.

3. *Dynamic interpretation/explanation of experimental results:* The results here are interesting – but lack a fair bit of interpretation from a PV framework regarding why the responses are occurring. The authors make some efforts on this front, but more needs to be done beyond just reporting the results to really enhance the impact of this study. For example, experiment 4 (changing the intensity of the PV anomaly) shows almost no change in response despite a presumably stronger PV gradient (though it might not be that much stronger given the experiment set-up). The results are interesting – but there's little to no interpretation for why we see the response we do. The same goes for experiment 5 – the authors report the change in tropospheric circulation but provide little interpretation for why. For example – why do we see a decrease in cyclonic relative vorticity with a widening PV anomaly? How can this be interpreted in a PV framework? Why is the surface circulation so much stronger?
We have tried to strengthen the dynamic interpretation of this work as the reviewer suggests. In experiment 4, we agree that is the lack of the change of gradient within the PV intrusion that is the likely cause of the lack of change in the circulation around the anomaly. However, we make

the argument that PV values within these intrusions are unlikely massive values far greater than that of the dynamical tropopause and therefore it reflects the real work atmosphere relatively well, despite this limit within the experimental design. In experiment 5, it is explained why we may see an increased relative vorticity value with broadened PV anomaly in the mid-tropopsphere.

**Minor comments:**
*General comments:*
- The manuscript reads *very* colloquially which is problematic. Please carefully check through the manuscript to identify instances where this occurs – I've identified some examples here, but there are many others throughout:
  - L40: Air can be advected or diabatically altered, but it cannot be 'introduced' to another region of the atmosphere.
    "introduced" changed to "advected"
  - L42-43: The term 'basic' here isn't necessary, and acts to undermine your study (there's little basic about PV theory – it's an advanced synoptic-dynamic topic that readers may not be familiar with).
    "Basic" has been removed as per suggestion
  - L67-80: You use the term 'This study examines/looks at/aims' too much here – aim to rework a bit.
    We have tried to lessen the use of these words as suggested
  - L92: 'diagnostic for reanalysis sets to diagnose' – aim to avoid repetitive words in a single sentence (there were several of these in the manuscript)
    Remove repeated words
  - L237-238 and elsewhere: Unless quantifying, avoid using the term 'stronger' and 'weaker' or similar qualitive statements (other examples include 'meagre' in L370 or 'massive' in L378)
    We have removed the use of qualitative words throughout the text as suggested.
  - L291 and elsewhere: The term 'exponentially' refers to a mathematically derived curved for a set of data points – if it is exponential, prove it; otherwise, please avoid statements that imply something different from what the data shows.
    References to exponential growth have been removed unless mathematically proved as suggested
  - L362 and elsewhere: Aim to avoid injecting opinion or emotion – lines such as 'Since we are dealing with …' should be avoided.
- Definition of terms: In several instances, terms/acronyms were introduced but not defined.  As a reader familiar with the topic, I could ascertain nearly everything, but less familiar readers may struggle.  Examples include (but aren't limited to):
  - PVU (L37) Updated: PVU ($1\ PVU = 10^6 K\ m^2\ s^{-1}\ kg^{-1}$)
  - COL (L64) COL = cut-off low. First instance has been updated.
  - Reference state (L86) – be sure to define what this is and how you establish it
  - "halo" (L150) A halo is a ring of light that encircles something bright. It is our best word to describe our anomaly setup
  - MSLP (L201) – MSLP = mean sea level pressure. First instance has been updated
  - Sphere of influence (L271) - removed
  - Mid-tropospheric (L347)
  - Total atmospheric system (L478) - removed
- Figures/equations: There were several inconsistencies in the figures that could be tightened up, and equation 6 does not need to be there (it's just a re-arrangement of equation 1 and can be stated as such):
  - Figure 3 and elsewhere:  Please always use panel labels (eg. A and B)

consistently. Please also include reference vectors whenever showing vectors that represent direction and magnitude.

We are not sure of the consistency the reviewer is looking for in the figure labels. We have used A and B consistently (A, B, A1, B1, etc) throughout the text

o Figure 5: This would be more clearly communicated as a table rather than a flow chart

We have converted this figure to a Table (1) as per the reviewers suggestion.

o Figures 9, 11, 14, and 16: Please use the same axis labels amongst these four figures. This is particularly important for Figure 14, which appears to have a large MSLP response based on figures 9, 11, and 16, but in reality is only ~0.3 hPa.

These have been updated as per the reviewers suggestion

o All captions: Be sure to include all relevant information, such as MSLP contour intervals or wind speed contour intervals.

Additional information such as MSLP and wind velocity contours are provided as per the suggestion

---

## Author Comment (AC2)

**"Stratospheric intrusion depth and its effect on surface cyclogenesis: An idealized PV inversion experiment"**

**Authors:** Barnes, Ndarana, Sprenger, and Landman

**Recommendation:** Major Revision

**Overview:**

In this study, the authors perform a series of idealized experiments in which they invert QGPV anomalies of various sizes, shapes, and vertical depths for their associated horizontal circulations. These circulations are then used to identify QGPV configurations that are likely to be more influential on surface cyclogenesis. In my opinion, while the analysis does not necessarily offer any new qualitative dynamical insight beyond what has already been garnered from the application of a PV framework in prior work, the experiments performed herein do provide a nice systematic quantitative treatment of how nuances in the structure of QGPV anomalies contribute to surface cyclogenesis. This quantitative assessment is novel from my perspective, and justifies the value of this study.

That being said, there several instances within the text in which I felt the present work could be better motivated and described with improved precision. Additionally, the intensification rate of surface cyclones is an important component of the interaction between upper-level PV anomalies and the surface. The temporal evolution of surface cyclones is not considered as part of the analysis but is quantifiable using diagnostic PV tendency inversions. Last, the analysis does not necessarily consider the role that static stability plays in modulating the character of these interactions and may represent an additional experiment that the authors can consider integrating into their analyses. Given the extent of my comments below, I have recommended the manuscript be returned for Major Revisions.

**Major Comments:**

**1.** There were several instances within the text in which I found myself a bit confused regarding the interpretation of figures (see minor comments below). I believe that this confusion could be remedied with a thorough review of the text to improve the precision of the discussion and through better definition of various terms. For example, it was difficult for me to differentiate between the physical interpretation of the minimum relative vorticity and the cross-sectional minimum in relative vorticity.

Thank you for this suggestion. We have made additions to the figures, text and captions as specified in the relevant suggestion below.

**2.** The introduction and motivation for the present work could be made clearer. Namely, it might be effective to construct a figure that highlights the diversity of PV intrusions and how these structures are associated with different surface cyclone intensities in real data. This figure could more effectively frame the idealized experiments performed in this study. There are also several instances in the introduction where the authors emphasize that such a study has not been performed in the Southern Hemisphere. But, to my knowledge, there is no reason to expect that PV anomalies will behave in a dynamically different way compared to the

Northern Hemisphere. Therefore, I recommend this discussion should be either more strongly motivated or eliminated from the text.

The reviewer makes a very valid suggestion. The work was inspired by two papers preceding this work – one of a case study of PV induced COLs and surface lows and another of a climatology of COL extensions associated with PV intrusion depths. Both are cited in this work. It is felt that this serves as justification for the diversity of PV intrusion scenarios tested in this study. Although real world cases could additionally be shown here, this may take away from the idealized feel of the work and make it rather lengthy. We appreciate this suggestion, but have opted to not include at this stage.

**3.** The authors do not perform a temporal diagnosis of the evolution/development of cyclonic circulations at the surface in their idealized experiments, but such an analysis can be performed in a diagnostic sense using either QGPV (e.g., Breeden and Martin 2018) or Ertel PV (e.g., Davis and Emanuel 1991). I wonder whether the application of this diagnostic framework for examining the instantaneous intensification rate of surface cyclones would bolster the analysis. It is also not clear in the text why QGPV is adopted over Ertel PV. At the very least, this choice should be justified in the context of the proposed applications.

The reviewer proposes a very interesting question regarding temporal diagnoses. The idealized setup reflects instantaneous changes of the parameters that are induced by the various PV anomalies. Although we did consider including temporal aspects visa case studies as was performed in Breeden and Martin (2018), it was decided that this would confuse the direction of the work. We therefore decided to the work limited to instantons changes and analysis. We hope that future work will entail the utilization of a dynamical core etc which we can then use to analyze temporal changes such as those proposed by the reviewer.

**4.** One element that is not explicitly considered in the idealized experiments is the role of static stability. Namely, those PV anomalies that are situated in a less stable environment are able to more effectively induce cyclogenesis. Could an experiment be run that considers varying the static stability of the environment? Additionally, the inversion of QGPV requires the specification of a reference atmosphere. It is not clear from the text what the authors have selected as their reference atmosphere, unless I may have missed it.

The reference atmosphere is described in the experimental setup section. The analysis of varying static stability environments was not considered here. We would however like to do this in future more intensely by looking at various aspects of the stability of the environments including low-level inversions etc. It believed that this could be best achieved by use of a full dynamical core which would also include temporal aspects of the stability of the environment (for eg. How is a low-level inversion degraded by PV anomaly?)

**5.** It is somewhat difficult to compare the various experiments because the summary figures (e.g., Fig. 9) feature different values along their y-axes. To better enable a comparison between experiments, I'd recommend standardizing these y-axes across all similar plots.

Thank you for this suggestion. We have standardized all of the y-axes in the plots specified.

**Minor, Specific, and Typographical Comments:**

*Abstract*

L15–16: I am a bit confused by the discussion in these lines. Namely, L15 states that horizontal extent is more important, whereas the next line states that vertical depth is important in dictating the strength of the circulation – both of which can be used to characterize cyclone intensity. Could these lines be clarified to better describe the respective influences of the vertical and horizontal extent of the stratospheric PV anomalies?

L17–18: This relationship in this sentence has to be true by definition, and I wonder whether it could be deleted to make more room to better clarify the nature of the relationships described in L15–16.

The abstract has been reconfigured to clarify the above as suggested. In essence, wider anomalies result in a greater decrease whilst relative vorticity on increases marginal whilst the reverse is true for surface relative vorticity changes.

*1. Introduction*

L41: The specification that "high PV" corresponds to negative values should occur earlier in the manuscript when this terminology is first used. I also believe it could be made clearer earlier in the introduction or abstract that the focus of the manuscript will be on Southern Hemispheric anomalies to avoid any potential confusion imparted on a reader.

The abstract has been amended in order to introduce the idea that the present study takes place in the Southern Hemisphere and introduce the idea that high-PV are negative anomalies in this study.

L44–45: The vertical depth of the circulation induced by the PV anomaly in Fig. 1 is also a function of the static stability. Namely, the lower static stability in the troposphere compared to the stratosphere allows the circulation induced by the PV anomaly to penetrate deeper towards the surface. Some reference to the thermodynamic environment in which the PV anomaly is embedded could benefit the discussion at this juncture in the text.

The static stability settings used within this study are added into the experimental setup section to reference the thermodynamic environment used within this study

L51: In the context of this study, cyclogenesis is described as a near-surface phenomenon. Consequently, I found the reference to cyclogenesis occurring throughout the stratosphere to be a bit confusing. Could this line be revised for improved clarity?

The use of cyclogenesis for the theoretical experimental framework and results have been removed which should clarify the lines referenced by the reviewer

L61: Why is it expected that the influence of PV anomalies will be different in the Southern Hemisphere compared to the Northern Hemisphere? I believe this claim may require stronger motivation/explanation.

The study in reference here is in relation to a climatology of PV intrusions in relation to COLs. The hemisphere in which it is performed is therefore highly relevant to the distribution of these phenomena etc.

The mathmathics and theories of the influence of PV anomalies is hemispheric independent and so for the purpose of this study may not be critical to the result obtained, as the reviewer suggests. We have therefore trimmed our references to it being a southern hemispheric study extensively. However, we have left some reference to this study being from a southern hemispheric perspective as we believe that this type of study in a SH context is rare. We hope

that the fact this study is performed from a SH point of view can help to stimulate further work in our region.

L62–63: The term "stratospheric tropopause" is not accurate, since the tropopause represents the interface between the troposphere and the stratosphere.
The typo has been removed and reference to "tropopause" only is made.

L64: The acronym, "COL", has not yet been defined in the manuscript.
The acronym COL has now been defined

L64–66: It is not clear to me how this conclusion follows from the previous sentences in the paragraph. Consider a revision to improve the clarity of the discussion.

The COL extension climatology by Barnes et al. (2021a) was based on real-case reanalysis data. As reanalysis data, climatological averages and composites were used, the isolated effect that the PV intrusions studied by Barnes et al. (2021a) had on surface cyclogenesis was not considered.

L72: As in L61, it is not clear why the physical influence of cyclonic PV anomalies will differ between the Northern and Southern Hemisphere simply because the sign of PV is negative in the Southern Hemisphere.
The mathemathics and theories of the influence of PV anomalies is hemispheric independent and so for the purpose of this study may not be critical to the result obtained, as the reviewer suggests. We have therefore trimmed our references to it being a southern hemispheric study extensively. However, we have left some reference to this study being from a southern hemispheric perspective as we believe that this type of study in a SH context is rare. We hope that the fact this study is performed from a SH point of view can help to stimulate further work in our region.

L72–73: This sentence is somewhat redundant with the statement that ends in L70. Consider whether it could be deleted.
We agree! The sentence has been deleted and incorporated into L70.

*2. Methodology*
L89: The Davis (1992) study focuses on the inversion of Ertel PV rather than QGPV, for which the system of equations for performing the inversion features nonlinear terms. For QGPV, the differential operator is linear, which does return a unique solution using successive over-relaxation.

L114–119: I found this discussion to be a bit confusing. Could you clarify more as to why the tropopause is defined differently within the inversion algorithm?
We have tried to reword this section in order to clarify this.

L121: Is the "specified DT" the height of the tropopause above ground level needed for the algorithm or the –1.5 PVU isosurface?
The specified DT is the height of the tropopause needed for the algorithm

L127: Is the intent to refer to the left panel of Fig. 3 in conjunction with this discussion? I ask

because the right panel of Fig. 3 does not show any pressure contours.
The reference to "(right)" has been removed. The intention was to discuss Fig. 3 holistically

L205–210: I found this discussion concerning vorticity thresholds to be a bit confusing. Are you basically looking for areas near the surface that feature vorticity with the magnitude described in the text? Or are you looking for areas where the circulation induced by the upper-level PV anomalies features vorticity of a particular magnitude at the surface.
L213–215: To what extent are the results sensitive to the selection of these vorticity thresholds?

The discussion concerning these cyclogenesis thresholds has been removed. The aim was to try and quantify the strength of cyclogenetic forcing but reviewers and readers found this comparison to cyclogenesis (with a temporal component implied) confusing. We have therefore removed this section

*3. Results*

L244: If referring to a line of constant pressure here and elsewhere in the manuscript, "isobar" is more descriptive than "isohypse".

Description of the surface pressure in terms of isobars has been removed as a result of our effort to separate the induced response and cyclogenesis type arguments.

L297–298: This particular sentence, as currently written, is a bit confusing. Would it be possible to rewrite it for further clarity? Similarly, I found L300–301 to also be confusing, which may require an edit for further clarity.

This description has changed as a result of our effort to separate the induced response and cyclogenesis type arguments.

L359–365: Could a plot of static stability be produced to help illustrate this effect more clearly?

L392: Arguably, this claim may be best reserved until after the final few experiments have been introduced (i.e., anomaly intensity has not been considered yet). Unless the goal here is to refer to the vertical extent of the anomaly. If so, a revision may be necessary to make that point clearer.

The intention was, as you refer, to refer to the vertical extent of the anomaly. We have tried to clarify this point with some minor wording tweaks.

The results in Experiments 1 and 2 imply that the proximity of a stratospheric intrusion to the surface has a larger impact on inducing deeper and enhanced cyclonic circulation at the surface than the vertical extent or size of the intrusion itself.

L406: Would it be possible to expand further on how this result may be an artifact of the basic state?

L421–427: I'm having a bit of a difficult time verifying some of these values against those plotted in Fig. 14. Could the text be revised to more clearly reference where these results are drawn from.

We have revised the text as suggested. In essence, there are minor differences in the meridional mid-tropospheric velocities but none really at the surface.

"Minimum cross-sectional relative vorticity decreases by $3 \times 10^{-5} s^{-1}$ from the low to high anomaly amplitude scenarios, whilst the maximum meridional velocity decreases by $1 m.s^{-1}$ around the anomaly. Anomalies of all magnitudes tested induce similar cyclogenetic forcing upon the surface. Both the induced surface pressure and relative vorticity are comparable throughout the scenarios tested."

L436–440: Could a figure be produced that shows the characteristic PV structure associated with these categories in real cases. This may help to visually motivate the forthcoming experiment.

L445–446: It is not clear what this particular statement is referring to.

For some reason our reference was left out of this sentence. This should clear up this issue.

L459–460: This statement is a bit difficult to verify. Namely, Fig. 16 suggests that the magnitude of the relative vorticity decreases with increasing width of the anomaly, which is counter to the discussion in this section. I think my confusion here may stem from difficulty understanding how the cross-sectional relative vorticity is calculated/defined.

We have tried to improve our arguments with respect to the lines indicated
"The intrusion width changes result in a change in the geometry of the resultant jet core which appears thinner and shorter the thinner anomaly, whilst the broader anomaly results in a visibly broader and longer jet core, affecting almost the entire cross-sectional domain. Although stronger velocities are observed in the troposphere as a result of broader intrusions, the mid-tropospheric relative vorticity increases sharply for broader intrusions (Figure 16). The larger magnitude relative vorticities induced by thinner intrusions are the result of the circulation, even though with lower velocity, being confined to a smaller horizontal region around the anomaly."

L462: For greater specificity, it may be worth stating that this 1 hPa corresponds to a pressure *perturbation* – as a "lower central pressure" would typically correspond to a stronger storm.
The line specified has been updated as per the suggestion in order to indicate that we are referring to a surface pressure anomaly

*4. Discussion and Conclusion*
L520–525: These few lines may be a bit redundant with the discussion in the previous paragraph. Consider whether these lines could be deleted without any loss of content.
In this paragraph we state that it is the height of the intrusion AGL that is more critical than the intrusion size itself. The previous paragraph talks to the depth experiments taking into account a similar tropopause height.

*Figures and Tables:*
Fig. 2: Would it be possible to specify the contour interval for PV in the caption?
Added as per reviewer suggestion

Fig. 5: It is not clear to me from the flow chart why all the experiments connect with the basic state box. It may be clearer to put the basic state box at the top of the image and then have all experiments flow beneath it.
As a result of this suggestion and that of the other reviewers we have converted this flow chart to a Table in order to improve how the variation of these experiments reads.

Fig. 6: Could the contour interval for the meridional wind be included as part of the caption?

Fig. 9: It is not clear to me how the cross-sectional minimum relative vorticity is different from the minimum relative vorticity value. Could this be explained a bit more clearly in the text?

Fig. 9/11/13: Would it be possible to make the limits along the y-axes the same in all of these plots so as to allow for better comparison between experiments?

References:

Breeden, M., and J. E. Martin, 2018: Analysis of the initiation of an extreme North Pacific jet retraction using piecewise tendency diagnosis. *Quart. J. Roy. Meteor. Soc.*, **144**, 1895–1913, https://doi.org/10.1002/qj.3388.

Davis, C. A., and K. A. Emanuel, 1991: Potential vorticity diagnostics of cyclogenesis. *Mon. Wea. Rev.*, **119**, 1929–1953, https://doi.org/10.1175/1520-0493(1991)119<1929:PVDOC>2.0.CO;2.

---

## Referee Report (RR1)

**Review**

**Stratospheric intrusion depth and its effect on surface cyclogenesis: An idealized PV inversion experiment**

by Michael A. Barnes, Thando Ndarana, Michael Sprenger and Willem A. Landman

**Summary**

The authors investigate the impact of stratospheric intrusions on the flow/vorticity and pressure field on the surface by quasi-geostrophic PV inversion. The authors perform several sensitivity experiments to look at the contribution of different parameters like intrusion depth and anomaly scale on the surface response.

The results are not surprising but might be nice to have in a kind of summary of different parameters. However, the manuscript is quite lengthy and written colloquially. I miss a thorough context of recent work in the introduction and the conclusions and a dynamical explanation of the experiments next to pure observational descriptions of the results. Additionally, I miss a thorough revision of the manuscript regarding some of the major concerns of both reviewers. Therefore, I still recommend major revisions before acceptance to meet the high standard of the journal.

**Major comments**

1.  Response to first review
    The authors should respond more carefully to both reviewers. Both reviewers made perfectly clear that there have been gaps in the interpretation and especially the presentation of the results, but only the few basic examples of the reviewers have been modified. The authors should check very carefully:

    a)  the colloquial writing (terms like „amount of surface cyclogenetic forcing" - throughout the text, „development driver" - L461, „drive stratospheric air into troposphere" - L407) is problematic and often leads to a difficult or even wrong interpretation (e.g. „High-PV anomalies of stratospheric air are often advected into the troposphere by Rossby wave breaking„ - L38)

    b)  interpretation of results: the authors mention in their response to reviewer 1 that a dynamical explanation is given following Experiment 5. But I cannot find any. The same holds true for the other experiments.

    c)  title: following the major comment of reviewer 1, I suggest to replace cyclogenesis in the titel, since just the cyclonic response of the stratospheric intrusion is investigated and no cyclogenesis

2.  Recent work
    The authors start the introduction directly with the properties of potential vorticity and the advantage of potential vorticity and how the flow around stratospheric intrusions looks like.. I would certainly argue that this paragraph could be shortened to a great deal. What I completely miss is a discussion about the importance of stratospheric intrusions in relation to cyclogenesis and

cyclones to motivate the current study further. What about the difference between baroclinic and barotropic cyclogenesis? This discussion would also help to motivate the use of a barotropic model. This point also holds for the discussion. The only studies the author relate to are the first authors last studies. I do really miss a more detailed context within the manuscript.

**Minor comments**

General comment

Especially in the introduction the manuscript is quite confusing regarding the northern and southern hemisphere since both perspectives are used. It would certainly improve the understanding, if the authors make clear at the beginning (not only in the abstract) that their work focus on southern hemispheric PV and then stick to it. That is, a stratospheric streamer on the southern hemisphere is in my understanding a low-PV anomaly or negative anomaly, not a high-PV (negative) anomaly (e.g. L14).

Detailed comments

L13: Do you refer to the strength of the cyclonic flow? What do you mean by amount?

L14: high-PV (negative) anomaly, s. above

L23: add e.g. before reference of (Røsting and Kristjánsson 2012)

L32: add e.g. before reference of (Davis and Emanuel 1991)

L34: Remove „the" before based on

L36: add e.g. before Lackmann 2011

L38: high PV anomalies -> low (SH)

L38: high PV anomalies are often advected into troposphere: pure advection would not lead to a mixture of stratospheric and tropospheric air without the influence of nonconservative processes as e.g. latent heat release, radiation or turbulence/mixing.

L40: can? In my opinion a positive anomaly (NH) is always associated with cyclonic flow

L41: has been -> have been

L42: has -> have

L57: few studies have used -> which?

L61: „The results show that stratospheric intrusions with a -1.5PVU tropopause associated with 250hPa COLs that extend to 300hPa or below, are more likely to result in surface cyclogenesis". - What are 250hPa COLs?

L71: again, what do you mean by amount?

L90: not integrating.. inverting!

L91ff: Davis 1992 investigates PV inversions under non-linear balance.. due to the non-linearity of the equations it is not expected that the resulting variables after piecewise inversion add to the full fields. However, you are considering PV inversion under quasi-geostrophic balance, that means the resulting variables add up to the full field. Hence, no sensitivity is expected how the inversion is performed.

Eq(3), Eq(4): I think in both equations a minus sign is missing

L407ff: reference to figures wrong.. -> Fig. 14 and 15 and not 15 and 16.

L459: „show the development of an amplifying trough", where do you show that? no development can be observed without time evolutions investigated

L532: Barnes et al. 2021 referenced twice

Figures and tables

Fig1: * remove one represent in caption

        * think -> thick

Can figures 2,3, and 4 combined to shorten manuscript? All figures really necessary?

Table 1: * Experiment 5 - typo 100km-100km?

        * Experiment 4 - either -1:-0.1:2 or -2:0.1:-1

---

## Referee Report (RR2)

**"Stratospheric intrusion depth and its effect on surface cyclogenesis: An idealized PV inversion experiment"**

**Authors:** Barnes, Ndarana, Sprenger, and Landman

**Recommendation:** Minor Revision

**Overview:**

In this study, the authors perform a series of idealized experiments in which they invert QGPV anomalies of various sizes, shapes, and vertical depths for their associated horizontal circulations. These circulations are then used to identify QGPV configurations that are likely to be more influential on surface cyclogenesis. The authors have done well to address my prior comments on the manuscript. At the moment, the large share of my comments are textual in nature, and offered to help improve the clarity and precision of the discussion. I have certainly found the results to be interesting, and believe that the manuscript will be ready for publication after a round of minor revision.

**Minor, Specific, and Typographical Comments:**

*1. Introduction*
L22: Rather than "down to", consider "drawn from" as a potential substitution in the text.

L26: I think the ending of this sentence is a bit unclear. Is this sentence referring to the ideas of PV invertibility or quasi-geostrophic theory? Consider a revision that improves the clarity of this sentence.

L49: The first two sentences of this paragraph are a bit redundant. Consider a revision to streamline the text a bit more.

L52–54: I find this particular sentence to be a bit vague and confusing. Could it be rewritten for improved clarity? I view this sentence to be important for setting the stage for the forthcoming analyses.

L59: I am still not sure why it should be expected that the effect should be different in the Southern Hemisphere versus the Northern Hemisphere, especially in an idealized environment. Consider corroborating the relevance of this statement more or simply keep the focus of the study on systematically examining the characteristics of these intrusions from an idealized perspective.

*2. Methodology*
L181: Consider revising the text to read as, "…is comprised of…", for greater clarity.

Overall: I love the table to summarize the various experiments – a great resource while evaluating the results.

*3. Results*
L212–213: I view this as a bit of a "chicken or the egg" type of description. I'm not sure I'm comfortable with saying that COLs are generated by stratospheric intrusions of high PV, since they *are* stratospheric intrusions of high PV. The generating mechanism, then, is what *causes* the intrusion. Consider a revision to the text accordingly.

L216: I believe the figure reference should be to Fig. 6b in this line, rather than Fig. 7b. Figure references appear to be off by 1 in many instances after this point in the manuscript.

L221: This quantity should be negative since we're in the Southern Hemisphere.

L224: Consider referencing any specific figures from this prior work that may help direct the reader to better verify this connection.

L239: I find the figure referencese to be unconventional. Consider using a more standard a,b,c,d,e,etc. label for panels rather than mixing numbers and letters multiple times (i.e., avoid 7b2 and stick with 7a, 7b, etc.).

L292: For additional clarity, it may be worthwhile to emphasize that this text refers to the austral summer.

L330: At the same time, the reduced tropospheric static stability in the high tropopause case can also allow for the circulation induced by the upper-level PV anomaly to penetrate to lower altitudes. I wonder if this is why you still see an effect of a lowered surface relative vorticity in Fig. 10 for the high tropopause cases, but its muted due to the competing effects between the penetration depth of the circulation and the height of the anomaly?

L341–342: These two sentences are a bit repetitive, could one be deleted for improved concision.

L369: This is largely semantics, but I view the use of the word, "intensity" throughout the manuscript to be a bit confusing. Namely, when I see intensity I instantly think "magnitude", but here the discussion refers to radial depth. Consider reviewing the text in the manuscript to improve the precision with which these changes to the PV anomalies are described. For example, in L380, "magnitude intensity" are the same words, from my perspective. Could it be possible to just keep reference to intensity in terms of anomaly magnitude and use radial depth or vertical depth to refer to changes in the anomaly's geometry?

L424–425: I believe this sentence refers to the wider PV anomaly, correct? If so, consider a revision to the sentence clarifying this point.

L434: Typo: One "shallower" should be removed from this sentence.

L438–441: Consider providing a physical explanation for this (i.e., the horizontal scale of the surface pressure distribution is larger for the broader anomalies, and thus the pressure gradient does not necessarily get much stronger as you increase the radial width of the anomaly).

*4. Discussion and Conclusions*
L478: Consider emphasizing that you are referring to the height above ground level in this sentence to promote further clarity.

L516: Consider adding the word, "environment", after baroclinic in this sentence.

*Figures and Tables:*
Fig. 8: I remain a bit confused as to what the difference between minimum relative vorticity and minimum cross-sectional relative vorticity is. Could this difference be more clearly identified within the body of the text before this first figure is introduced? Apologies if it is described earlier in the text and I missed it.

---

## Referee Report (RR3)

Review 3

**Stratospheric intrusion depth and its effect on surface cyclogenetic forcing: An idealized PV inversion experiment**

by Barnes et al.

The manuscript improved during the last revision. However, I still do have suggestions to improve the manuscript. I think after this round of minor revisions the manuscript might be ready for publication.

**Penetration depth**

The authors argue in Experiment 2 in LL454 that a lower tropopause results in an increased static stability leading to increased rotation (decreasing negative vorticity) around the anomaly. However, the penetration depth H of an anomaly varies inversely with the ambient static stability N around an anomaly (e.g. Martin 2013):

H = f L/N with L the horizontal scale of the anomaly.                    (1)

I would ask the authors to insert potential temperature contours especially into Fig9 and Fig14 to confirm their argument and solve this contradiction. I would expect to see a lower static stability in cases with lower tropopause height since the cyclonic flow is stronger on the surface.
In Experiment 5 the penetration depth nicely varies with horizontal scale as expected following equation (1). I recommend to include the well-known concept of penetrations depth into the discussion of both, experiment 2 and 5.

**Specific and technical corrections**

L13: high PV (large negative PV) -> remove information about negative PV anomalies in abstract or mention focus on southern hemisphere.. otherwise confusing

L22: add that after fact: is the fact **that** PV

L60: Bierly confirm**ed**/show**ed**

L128-L130: piecewise PV inversion is no methodology of solving equation (1), but more an extended approach thanks to the additive behaviour of PV anomalies. Furthermore the different techniques studied by Davis 1992 are only necessary for PV inversion under nonlinear balance (since the equations are nonlinear) and not quasi-geostrophic balance. Under quasi-geostrophic balance the flow fields from different piecewise inversion are additive and do not depend on the different approaches suggested by Davis 1992. I hence suggest to remove this paragraph.

L140: remove „using a piecewise numerical approach"

L157: define AGL the first time using it

Eq5: x-xi -> x-x_pos

L419: systemic -> systematic?

L519: lesser -> less

L551: ie. By -> i.e. by
L573: ie. That -> i.e. that
L607: result -> results
L609: why although? there is no contradiction, is it?

LL610: The authors state that „The larger magnitude relative vorticities induced by thinner intrusions are the result of the circulation with lower velocity being confined to a smaller horizontal region around the anomaly." However, from Fig14 I identify larger magnitude in relative vorticity for broad intrusions.. the rel. vorticity is more negative for broad intrusions and hence the cyclonic circulation is stronger. Please clarify! Especially in L638 the authors write „Enhanced surface cyclonic rotation is also induced by the broader PV anomaly with increases in the surface relative vorticity. " and contradict themselves.

L670: ie. -> i.e.

L704: I would not state key finding, since all results are not knew but nice to have summarized in one study.

L715: remove of before with

**References**

Martin, Jonathan E. *Mid-latitude atmospheric dynamics: a first course*. John Wiley & Sons, 2013.

---

## Author Response (AR2)

**Review**

**Stratospheric intrusion depth and its effect on surface cyclogenesis: An idealized PV inversion experiment**

by Michael A. Barnes, Thando Ndarana, Michael Sprenger and Willem A. Landman

**Summary**

The authors investigate the impact of stratospheric intrusions on the flow/vorticity and pressure field on the surface by quasi-geostrophic PV inversion. The authors perform several sensitivity experiments to look at the contribution of different parameters like intrusion depth and anomaly scale on the surface response.

The results are not surprising but might be nice to have in a kind of summary of different parameters. However, the manuscript is quite lengthy and written colloquially. I miss a thorough context of recent work in the introduction and the conclusions and a dynamical explanation of the experiments next to pure observational descriptions of the results. Additionally, I miss a thorough revision of the manuscript regarding some of the major concerns of both reviewers. Therefore, I still recommend major revisions before acceptance to meet the high standard of the journal.

**Major comments**

1. Response to first review
   The authors should respond more carefully to both reviewers. Both reviewers made perfectly clear that there have been gaps in the interpretation and especially the presentation of the results, but only the few basic examples of the reviewers have been modified. The authors should check very carefully:

a) the colloquial writing (terms like „amount of surface cyclogenetic forcing" - throughout the text, „development driver" - L461, „drive stratospheric air into troposphere" - L407) is problematic and often leads to a difficult or even wrong interpretation (e.g. „High-PV anomalies of stratospheric air are often advected into the troposphere by Rossby wave breaking„ - L38)
   Amount of: Surface cyclogenetic forcing is measured by means of changes in the induced relative vorticity and surface pressure for each intrusion scenario.
   L407: RWB events, which are associated with isentropic transport of stratospheric air
   L461: This re-emphasises that upper-level processes induce both the surface cyclone and
   L38: Rossby wave breaking (RWB) is often associated with the isentropic transport of high-PV (large negative values in the Southern Hemisphere) anomalies of stratospheric air into the troposphere

b) interpretation of results: the authors mention in their response to reviewer 1 that a dynamical explanation is given following Experiment 5. But I cannot find any. The same holds true for the other experiments.
   This study links the vertical geometries of idealized PV intrusions to its affect on surface cyclogenetic forcing. We discuss these geometries in terms of different weather systems. For example, in Experiment 5 where we link our experimentation with PV intrusion width with different types of RWB events and for example in Experiment 2-3 with increased tropopause heights in a tropical/summer scenario. We feel that this adequately interprets the results in terms of observed weather systems.

c) title: following the major comment of reviewer 1, I suggest to replace cyclogenesis in the titel, since just the cyclonic response of the stratospheric intrusion is investigated and no cyclogenesis
   The title has been changed as per the reviewer's suggestion.

2. Recent work
   The authors start the introduction directly with the properties of potential vorticity and the

advantage of potential vorticity and how the flow around stratospheric intrusions looks like.. I would certainly argue that this paragraph could be shortened to a great deal. What I completely miss is a discussion about the importance of stratospheric intrusions in relation to cyclogenesis and cyclones to motivate the current study further. What about the difference between baroclinic and barotropic cyclogenesis? This discussion would also help to motivate the use of a barotropic model. This point also holds for the discussion. The only studies the author relate to are the first authors last studies. I do really miss a more detailed context within the manuscript.

The presence of high-PV intrusion of stratospheric air in terms of surface cyclogenesis are well established. The authors have tried to motivate this study as per the reviewers suggestion by pointing out a variety of studies that show this in fact to be the case:

"Several studies have shown cases of cyclogenesis and their development in the presence of a stratospheric intrusions of high-PV (eg. Davis and Emanuel 1991; Davis 1992a; Iwabe and Da Rocha 2009; Barnes et al. 2021c). Bierly (1997) confirm this link through composite analysis and show the importance of the upper-level intrusion during cyclones initial development. Many studies have focussed on rapid cyclogenesis. A landmark case study shows a tropopause fold that developed in relation to the President's Day cyclone over the east coast of the United States (Uccellini et al. 1985). Rapid cyclogenesis has since been linked to the presence of a PV tower – an alignment of three distinct PV anomalies, in the upper troposphere, lower troposphere and surface (eg. Čampa and Wernli 2012)."

**Minor comments**

General comment

Especially in the introduction the manuscript is quite confusing regarding the northern and southern hemisphere since both perspectives are used. It would certainly improve the understanding, if the authors make clear at the beginning (not only in the abstract) that their work focus on southern hemispheric PV and then stick to it. That is, a stratospheric streamer on the southern hemisphere is in my understanding a low-PV anomaly or negative anomaly, not a high-PV (negative) anomaly (e.g. L14).
In this work, we deal with large negative PV values. We have avoided using the term "low-PV" values as tends to imply smaller (less intense) values. In order to solve this issue we clarify in the beginning of the manuscript that the convention of this work is to use high-PV values within the context of this study to indicate large absolute PV values and since we are working in the SH, we deal with large negative values as being "high-PV" values.

"It should be noted that this takes place in the southern hemisphere atmosphere where large negative values of PV are associated with cyclonic motion, contrary to the northern hemisphere where cyclonic motion is associated with large positive values. For the purposes of this study, high-PV values are associated with large negative values of PV."

Detailed comments

L13: Do you refer to the strength of the cyclonic flow? What do you mean by amount?
"Amount" has been changed by "intensity" to clarify
L14: high-PV (negative) anomaly, s. above
The "negative anomalous" has been changed to "large negative PV" to avoid confusion.
L23: add e.g. before reference of (Røsting and Kristjánsson 2012)
Changed as per suggestion
L32: add e.g. before reference of (Davis and Emanuel 1991)
Changed as per suggestion
L34: Remove „the" before based on
Changed as per suggestion
L36: add e.g. before Lackmann 2011
Changed as per suggestion
L38: high PV anomalies -> low (SH)
Brackets "(large negative values in the Southern Hemisphere)" have been used to clarify our meaning

of "high-PV".

L38: high PV anomalies are often advected into troposphere: pure advection would not lead to a mixture of stratospheric and tropospheric air without the influence of nonconservative processes as e.g. latent heat release, radiation or turbulence/mixing.

Changed the word "advected" to "transported" to avoid confusion.

L40: can? In my opinion a positive anomaly (NH) is always associated with cyclonic flow

The word "can" has been removed.

L41: has been -> have been

Changed as per suggestion

L42: has -> have

Changed as per suggestion

L57: few studies have used -> which?

L61: „The results show that stratospheric intrusions with a -1.5PVU tropopause associated with 250hPa COLs that extend to 300hPa or below, are more likely to result in surface cyclogenesis„. - What are 250hPa COLs?

In the context of the study mentioned, we refer to COLs at the 250hPa level. This has been clarified by changing the phrase to "COLs detected on the 250hPa pressure level"

L71: again, what do you mean by amount?

"Amount" has been changed by "intensity" to clarify

L90: not integrating.. inverting!

Changed as per suggestion

L91: Davis 1992 investigates PV inversions under non-linear balance.. due to the non-linearity of the equations it is not expected that the resulting variables after piecewise inversion add to the full fields. However, you are considering PV inversion under quasi-geostrophic balance, that means the resulting variables add up to the full field. Hence, no sensitivity is expected how the inversion is performed.

Eq(3), Eq(4): I think in both equations a minus sign is missing

Minus sign was missing in Equation 4. It has been rectified.

L407: reference to figures wrong.. -> Fig. 14 and 15 and not 15 and 16.

Unclear where the reviewer is referring. No reference to Fig 14 and 15 on L407. All references to Fig 14 and 15 look accurate.

L459: „show the development of an amplifying trough", where do you show that? no development can be observed without time evolutions investigated

The temporal factor of this has been removed to only indicate that the PV anomaly simulates upper-level trough as is expected by theory. "… show that the high-PV anomaly results in a trough, as is expected by theory"

L532: Barnes et al. 2021 referenced twice Figures and tables
Fig1: * remove one represent in caption

Changed as per suggestion

* think -> thick

Changed as per suggestion

Can figures 2,3, and 4 combined to shorten manuscript? All figures really necessary?

We feel that all are necessary as they link to the experimental figures in the below sections directly.

Table 1: * Experiment 5 - typo 100km-100km?

We refer here to 100km-800km in steps of 100km. No changes made.

* Experiment 4 - either -1:-0.1:2 or -2:0.1:-1

Changed as per suggestion

**"Stratospheric intrusion depth and its effect on surface cyclogenesis: An idealized PV inversion experiment"**

**Authors:** Barnes, Ndarana, Sprenger, and Landman

**Recommendation:** Minor Revision

**Overview:**

In this study, the authors perform a series of idealized experiments in which they invert QGPV anomalies of various sizes, shapes, and vertical depths for their associated horizontal circulations. These circulations are then used to identify QGPV configurations that are likely tobe more influential on surface cyclogenesis. The authors have done well to address my prior comments on the manuscript. At the moment, the large share of my comments are textual in nature, and offered to help improve the clarity and precision of the discussion. I have certainly found the results to be interesting, and believe that the manuscript will be ready for publicationafter a round of minor revision.

**Minor, Specific, and Typographical Comments:**

*1. Introduction*
L22: Rather than "down to", consider "drawn from" as a potential substitution in the text.
Changed applied as per reviewer suggestion

L26: I think the ending of this sentence is a bit unclear. Is this sentence referring to the ideas of PV invertibility or quasi-geostrophic theory? Consider a revision that improves the clarity of this sentence.
Sentence restructured to emphasize we are referring to the ideas of PV invertibility:
"PV invertibility became more refined and generalised through the development of quasi-geostrophic theory (Charney and Stern 1962) and is still continually being developed and improved on today."

L49: The first two sentences of this paragraph are a bit redundant. Consider a revision to streamline the text a bit more.
Sentences have been restructured as suggested.

L52–54: I find this particular sentence to be a bit vague and confusing. Could it be rewritten for improved clarity? I view this sentence to be important for setting the stage for the forthcoming analyses.
Sentences have been restructured as suggested.

L59: I am still not sure why it should be expected that the effect should be different in the Southern Hemisphere versus the Northern Hemisphere, especially in an idealized environment. Consider corroborating the relevance of this statement more or simply keep the focus of the study on systematically examining the characteristics of these intrusions from an idealized perspective.
We agree that from a mathematical stand-point that the hemisphere in which we are working may be insignificant. However, from the perspective of science in the global south, we feel that this study is significant as most of the work has been done from a northern hemispheric perspective. We would therefore like to keep this distinction in this work, even if this point is made subtly

*2. Methodology*
L181: Consider revising the text to read as, "…is comprised of…", for greater clarity.
Sentences have been restructured as suggested.

Overall: I love the table to summarize the various experiments – a great resource while evaluating the results.

*3. Results*

L212–213: I view this as a bit of a "chicken or the egg" type of description. I'm not sure I'm comfortable with saying that COLs are generated by stratospheric intrusions of high PV, since they *are* stratospheric intrusions of high PV. The generating mechanism, then, is what *causes* the intrusion. Consider a revision to the text accordingly.

The concern regarding the wording is understandable. We have changed the phrase suggesting COLs are "generated by" stratospheric intrusions to "associated with"

L216: I believe the figure reference should be to Fig. 6b in this line, rather than Fig. 7b. Figure references appear to be off by 1 in many instances after this point in the manuscript.

Corrected. There was an issue with figure references which have been corrected.

L221: This quantity should be negative since we're in the Southern Hemisphere.

Correction applied as per suggestion.

L224: Consider referencing any specific figures from this prior work that may help direct the reader to better verify this connection.

We have added in a reference to the Table in this previous work where this is shown.

L239: I find the figure referencese to be unconventional. Consider using a more standard a,b,c,d,e,etc. label for panels rather than mixing numbers and letters multiple times (i.e., avoid 7b2 and stick with 7a, 7b, etc.).

The naming convention we have used references the p[art of the figure we are referring to. Ie. Here we refer to column B, row 2 of Figure 7 (Figure 7-B2). To clarify this naming convention we have included a description of this convention in the first instance of this (Fig 8) "By convention, in-text figure references refer to the column letter and the row number of the panel referenced (ie B2 refers to the panel in column B and row 2)"

L292: For additional clarity, it may be worthwhile to emphasize that this text refers to the austral summer.

Change applied as suggested

L330: At the same time, the reduced tropospheric static stability in the high tropopause case canalso allow for the circulation induced by the upper-level PV anomaly to penetrate to lower altitudes. I wonder if this is why you still see an effect of a lowered surface relative vorticity in Fig. 10 for the high tropopause cases, but its muted due to the competing effects between the penetration depth of the circulation and the height of the anomaly?

L341–342: These two sentences are a bit repetitive, could one be deleted for improved concision.

The sentence has been combine and reduced as suggested:

"Enhanced cyclonic circulation is induced at the surface in the lower tropopause scenario as shown by an increase in the amount of cyclonic surface relative vorticity."

L369: This is largely semantics, but I view the use of the word, "intensity" throughout the manuscript to be a bit confusing. Namely, when I see intensity I instantly think "magnitude", but here the discussion refers to radial depth. Consider reviewing the text in the manuscript to improve the precision with which these changes to the PV anomalies are described. For example,in L380, "magnitude intensity" are the same words, from my perspective. Could it be possible tojust keep reference to intensity in terms of anomaly magnitude and use radial depth or vertical depth to refer to changes in the anomaly's geometry?

The confusion of the different "intensities" is very well noted. The word depth is however also

ambiguous in this work since we also refer to depth with regards to the intrusion's "reach" towards the surface. In order to clarify we have changed our references to "vertical intensity" referring to the increased radial depth by referring to "vertical extent" or "radial height" or both.

L424–425: I believe this sentence refers to the wider PV anomaly, correct? If so, consider a revision to the sentence clarifying this point.
Correct. Sentence was incomplete. It has been rectified.
"Conversely, the wider intrusion results in an increase in the maximum mid-tropospheric meridional velocities compared to the standard configuration ($15m.s^{-1}$ compared to $11m.s^{-1}$ in Experiment 0)."

L434: Typo: One "shallower" should be removed from this sentence.
Change applied as per suggestion

L438–441: Consider providing a physical explanation for this (i.e., the horizontal scale of the surface pressure distribution is larger for the broader anomalies, and thus the pressure gradient does not necessarily get much stronger as you increase the radial width of the anomaly).
A physical description similar to that offered by the reviewer has been added.

*4. Discussion and Conclusions*
L478: Consider emphasizing that you are referring to the height above ground level in this sentence to promote further clarity.
Height AGL has been specified as per suggestion

L516: Consider adding the word, "environment", after baroclinic in this sentence.
Change applied as per suggestion

*Figures and Tables:*
Fig. 8: I remain a bit confused as to what the difference between minimum relative vorticity and minimum cross-sectional relative vorticity is. Could this difference be more clearly identified within the body of the text before this first figure is introduced? Apologies if it is described earlier in the text and I missed it.
We have added in an explicit definition as per the reviewers suggestion at the first reference of minimum cross-sectional vorticity in the text.

---

## Author Response (AR3)

Review 3

**Stratospheric intrusion depth and its effect on surface cyclogenetic forcing: An idealized PV inversion experiment**

by Barnes et al.

The manuscript improved during the last revision. However, I still do have suggestions to improve the manuscript. I think after this round of minor revisions the manuscript might be ready for publication.

**Penetration depth**

The authors argue in Experiment 2 in LL454 that a lower tropopause results in an increased static stability leading to increased rotation (decreasing negative vorticity) around the anomaly. However, the penetration depth H of an anomaly varies inversely with the ambient static stability N around an anomaly (e.g. Martin 2013):

H = f L/N with L the horizontal scale of the anomaly.                                                   (1)

I would ask the authors to insert potential temperature contours especially into Fig9 and Fig14 to confirm their argument and solve this contradiction. I would expect to see a lower static stability in cases with lower tropopause height since the cyclonic flow is stronger on the surface.
In Experiment 5 the penetration depth nicely varies with horizontal scale as expected following equation (1). I recommend to include the well-known concept of penetrations depth into the discussion of both, experiment 2 and 5.

The contradiction is well noted. The ambient potential temperature gradient is constant in the troposphere throughout all the experiments as defined by the static stability settings. However, it is the local static stability (local potential temperature) that is enhance when the PV intrusion is closer to the surface. Ie. All values of the equation H=fL/N are equal in all three cases of Experiment 2 since H, L, f and N are all held constant.
We have aimed to clarify this contradiction argument by speaking more directly to the various terms in the QGPV equation in the paper in our arguments of Experiment 2.
In order to keep the figures clean, we have not shown the potential temperature contours as suggested, however we have quoted values of increased potential temperature gradients within the intrusion.

"

$$\zeta_g + f = q - f \frac{\partial}{\partial z}\left( \frac{\partial \bar{\theta}}{\partial z}^{-1} \theta \right)$$                                                   (1)

The environmental potential temperature gradient is set as constant since the tropospheric and stratospheric static stability values are kept constant throughout Experiment 2 (as explained in Section 2.2). Therefore, the term $\frac{\partial \bar{\theta}}{\partial z}^{-1}$ in Equation 6 does not change between the high and low tropopause cases. As a result, the change in relative vorticity is controlled by a local change in the potential temperature gradient ($\frac{\partial \theta}{\partial z}$). Calculation of this term within the intrusion shows that there is an increase in the local potential temperature gradient between three scenarios in Experiment 2. In fact, in the center of the intrusion at the height of the maximum meridional velocity, the potential temperature gradient ($\frac{\partial \theta}{\partial z}$) increases from the 0.44K. m$^{-1}$ in the higher tropopause scenario to 0.47K. m$^{-1}$ in lower tropopause scenario. Since $\frac{\partial}{\partial z}\left( \frac{\partial \bar{\theta}}{\partial z}^{-1} \right) < 0$ and both $q$ and $f$ are negative in the Southern Hemisphere, it follows that an increase in $\frac{\partial \theta}{\partial z}$ will result in a decrease in relative vorticity in the Southern Hemisphere. The more tightly packed potential temperature contours within the intrusion (higher static stability) in the scenario where the dynamical tropopause is closer to the surface results in a decreased cross-sectional relative vorticity compared to the higher tropopause scenario. "

**Specific and technical corrections**

L13: high PV (large negative PV) -> remove information about negative PV anomalies in abstract or mention focus on southern hemisphere.. otherwise confusing
*Removed negative reference in the abstract as per suggestion*

L22: add that after fact: is the fact **that** PV
*Change applied as per suggestion*

L60: Bierly confirm**ed**/show**ed**
*Change applied as per suggestion*

L128-L130: piecewise PV inversion is no methodology of solving equation (1), but more an extended approach thanks to the additive behaviour of PV anomalies. Furthermore the different techniques studied by Davis 1992 are only necessary for PV inversion under nonlinear balance (since the equations are nonlinear) and not quasi-geostrophic balance. Under quasi-geostrophic balance the flow fields from different piecewise inversion are additive and do not depend on the different approaches suggested by Davis 1992. I hence suggest to remove this paragraph.
*Change applied as per suggestion*

L140: remove „using a piecewise numerical approach"
*Changes applied as per suggestion*

L157: define AGL the first time using it
*AGL first used and defined on L119*

Eq5: x-xi -> x-x_pos
*Changes applied as per suggestion*

L419: systemic -> systematic?
*Changes applied as per suggestion*

L519: lesser -> less
*Changes applied as per suggestion*
L551: ie. By -> i.e. by *Changes applied as per suggestion*
*Changes applied as per suggestion*

L573: ie. That -> i.e. that
*Changes applied as per suggestion*

L607: result -> results
*Changes applied as per suggestion*

L609: why although? there is no contradiction, is it? (L445)

LL610: The authors state that „The larger magnitude relative vorticities induced by thinner intrusions are the result of the circulation with lower velocity being confined to a smaller horizontal region around the anomaly." However, from Fig14 I identify larger magnitude in relative vorticity for broad intrusions.. the rel. vorticity is more negative for broad intrusions and hence the cyclonic circulation is stronger. Please clarify! Especially in L638 the authors write „Enhanced surface cyclonic rotation is also induced by the broader PV anomaly with increases in the surface relative vorticity. " and contradict themselves.

*The first quotation referred to by the reviewer ("The larger magnitude relative vorticities induced by thinner intrusions are the result of the circulation with lower velocity being confined to a smaller horizontal region around the anomaly.") is in reference to the mid-tropospheric relative vorticity, whilst the second ("Enhanced surface cyclonic rotation is also induced by the broader PV anomaly with increases in the surface relative vorticity.") refers to the differences in relative vorticity at the surface. Relative vorticities are enhanced for thinner intrusions in the mid-troposphere whilst are limited at the surface, whilst the reverse is true for broader intrusions. We have clarified this by adapting the wording of this section to explicitly say which vertical levels we are referring to.*

L670: ie. -> i.e.
Changes applied as per suggestion

L704: I would not state key finding, since all results are not knew but nice to have summarized in one study.
Changed to "The relative contribution of different factors to surface cyclogenetic forcing is highlighted in this study."

L715: remove of before with

Changes applied as per suggestion

**References**

Martin, Jonathan E. *Mid-latitude atmospheric dynamics: a first course*. John Wiley & Sons